# Hygroscopic behavior and chemical composition evolution of internally mixed aerosols composed of oxalic acid and ammonium sulfate

Xiaowei Wang[1,2], Bo Jing[3], Fang Tan[3,4], Jiabi Ma[1], Yunhong Zhang[1] and Maofa Ge[3,4,5]

[1]The Institute of Chemical Physics, School of Chemistry and Chemical Engineering, Beijing Institute of Technology, Beijing 100081, P. R. China

[2]School of Chemical Engineering and Pharmaceutics, Henan University of Science and Technology, Luoyang 471023, P. R. China

[3]Beijing National Laboratory for Molecular Sciences (BNLMS), State Key Laboratory for Structural Chemistry of Unstable and Stable Species, CAS Research/Education Center for Excellence in Molecular Sciences, Institute of Chemistry, Chinese Academy of Sciences, Beijing 100190, P. R. China

[4]University of Chinese Academy of Sciences, Beijing 100049, P. R. China

[5]Center for Excellence in Regional Atmospheric Environment, Institute of Urban Environment, Chinese Academy of Sciences, Xiamen 361021, P. R. China

*Correspondence to:* Yunhong Zhang (yhz@bit.edu.cn) and Maofa Ge (gemaofa@iccas.ac.cn)

**Abstract**

Although water uptake of aerosol particles plays an important role in the atmospheric environment, the effects of interactions between components on chemical composition and hygroscopicity of particles are still not well constrained. The hygroscopic properties and phase transformation of oxalic acid (OA) and mixed particles composed of ammonium sulfate (AS) and OA with different organic to inorganic molar ratios (OIRs) have been investigated by using confocal Raman spectroscopy. It is found that OA droplets first crystallize to form oxalic acid dihydrate at 71% relative humidity (RH), and further lose crystalline water to convert into anhydrous oxalic acid around 5% RH during the dehydration process. The deliquescence and efflorescence point for AS is determined to be 80.1 $\pm$ 1.5% RH and 44.3 $\pm$ 2.5% RH, respectively. The observed efflorescence relative humidity (ERH) for mixed OA/AS droplets with OIRs of 1:3, 1:1 and 3:1 is 34.4 $\pm$ 2.0% RH, 44.3 $\pm$ 2.5% RH and 64.4 $\pm$ 3.0% RH, respectively, indicating the elevated OA content appears to favor the crystallization of mixed systems at higher RH. However, the deliquescence relative humidity (DRH) of AS in mixed OA/AS particles with an OIR of 1:3 and 1:1 is observed to occur at 81.1 $\pm$ 1.5% RH and 77 $\pm$ 1.0% RH, respectively. The Raman spectra of mixed OA/AS droplets indicate the formation of ammonium hydrogen oxalate ($NH_4HC_2O_4$) and ammonium hydrogen

sulfate ($NH_4HSO_4$) from interactions between OA and AS in aerosols during the dehydration process in the time scale of hours, which considerably influence the subsequent deliquescence behavior of internally mixed particles with different OIRs. The mixed OA/AS particles with an OIR of 3:1 exhibit no deliquescence transition over the RH range studied due to the considerable transformation of $(NH_4)_2SO_4$ into $NH_4HC_2O_4$ with a high DRH. Although the hygroscopic growth of mixed OA/AS droplets is comparable to that of AS or OA at high RH during the dehydration process, Raman growth factors of mixed particles after deliquescence are substantially lower than those of mixed OA/AS droplets during the efflorescence process and further decrease with elevated OA content. The discrepancies for Raman growth factors of mixed OA/AS particles between the dehydration and hydration process at high RH can be attributed to the significant formation of $NH_4HC_2O_4$ and residual OA, which remain solid at high RH and thus result in less water uptake of mixed particles. These findings improve the understanding of the role of reactions between dicarboxylic acid and inorganic salt in the chemical and physical properties of aerosol particles, and might have important implications for atmospheric chemistry.

**1 Introduction**

Atmospheric aerosols have vital impacts on the Earth's climate directly by scattering, reflecting and absorbing solar radiation, and indirectly by influencing formation of clouds and precipitation (Tang and Munkelwitz, 1994b; Jacobson et al., 2000; Penner et al., 2001; Pöschl, 2005; Martin, 2000; Von Schneidemesser et al., 2015). Direct and indirect effects depend on the chemical and physical properties of atmospheric aerosols, including size, structure, hygroscopicity and chemical composition. Field observations indicate that aerosol particles are generally internal mixtures of inorganic and organic compounds in the atmosphere (Saxena et al., 1995; Murphy et al., 1998; Murphy et al., 2006; Pratt and Prather, 2010). Ammonium sulfate (AS) is one of the most abundant inorganic constituents in the atmosphere, hygroscopicity of which has been widely investigated (Liu et al., 2008; Cziczo et al., 1997; Laskina et al., 2015).

Oxalic acid (OA) is ubiquitous and has been identified as the dominant dicarboxylic acid in urban and remote atmospheric aerosols (Chebbi and Carlier, 1996; Kanakidou et al., 2004; Yang and Yu, 2008; Wang et al., 2012; Kawamura and Bikkina, 2016). Previous studies have focused on deliquescence behavior of pure OA (Peng et al., 2001; Braban et al., 2003; Miñambres et al., 2013; Ma et al., 2013a; Jing et al., 2016). It was found that due to its high deliquescence point OA

exhibited no deliquescence transition or hygroscopic growth within relative humidity (RH) range studied by an electrodynamic balance (EDB) (Peng et al., 2001), vapor sorption analyzer (Ma et al., 2013a) or hygroscopicity tandem differential mobility analyzer (HTDMA) (Jing et al., 2016). Braban et al. (2003) reported that OA could deliquesce at 98% RH using aerosol flow tube Fourier transform infrared spectroscopy (AFT-FTIR). However, the study on the efflorescence behavior of OA during the dehydration process remains limited (Peng et al., 2001; Mikhailov et al., 2009). Peng et al. (2001) observed the efflorescence transition of OA using EDB while Mikhailov et al. (2009) reported continuous hygroscopic growth of OA during both hydration and dehydration process using the HTDMA.

The dicarboxylic acids can affect properties of internally mixed aerosol particles such as hygroscopicity, phase transition, solubility and chemical reactivity (Lightstone et al., 2000; Brooks et al., 2002; Sjogren et al., 2007; Pradeep Kumar et al., 2003; Treuel et al., 2011; Laskin et al., 2012; Drozd et al., 2014; Peng et al., 2016; Jing et al., 2016; Li et al., 2017; Jing et al., 2017). Field measurements have observed the formation of low-volatility organic salts in atmospheric particles due to the reactions of organic acids with mineral salts, chloride salts, nitrate salts, ammonium and amines (Sullivan and Prather, 2007; Laskin et al., 2012; Wang and Laskin, 2014; Smith et al., 2010). The organic salts formed typically have varying hygroscopicity compared to the corresponding organic acids. Thus, these drastic changes in aerosol composition have potential impacts on the water uptake and related physicochemical properties of particles. The effects of OA on deliquescence behaviors of AS have been extensively investigated (Brooks et al., 2002; Prenni et al., 2003; Wise et al., 2003; Miñambres et al., 2013; Jing et al., 2016). The majority of studies found that the presence of OA had no obvious impacts on the deliquescence process of OA/AS mixtures with minor OA content (Brooks et al., 2002; Prenni et al., 2003; Wise et al., 2003). To our knowledge, there is still a lack of studies on the efflorescence process of OA/AS mixed systems. In fact, the efflorescence behavior is a critical hygroscopic characteristic of atmospheric aerosols, which may favor specific chemical interactions between components within the supersaturated droplets. For example, previous studies have found that the chloride depletion could occur in the NaCl/dicarboxylic acids mixed aerosols during the dehydration or efflorescence process, which led to the formation of organic salts and in turn affected subsequent deliquescence behaviors of aerosols (Laskin et al., 2012; Ghorai et al., 2014). Oxalic acid has been found to react with both mono- and

di-valent cations to form low volatility and solubility compounds (Drozd et al., 2014). Miñambres

et al. (2013) proposed that OA might react with AS to form ammonium hydrogen oxalate and

ammonium hydrogen sulfate within OA/AS solution. Due to the lack of available thermodynamic

data, the aerosol thermodynamic models typically assume that upon dehydration dicarboxylic acid

could only form organic solid without the organic salt in the inorganic electrolyte/dicarboxylic acid

system (Clegg and Seinfeld, 2006; Amundson et al., 2007). Thus, the incorporation of organic salts

formed from interactions between inorganic salts and organic acids is crucial in the modeling of

hygroscopic properties of mixed organic/inorganic particles. It merits further investigation on the

interactions between OA and AS and related influence on the water uptake behaviors of aerosols

during the dehydration and hydration processes.

Raman spectroscopy is a powerful technique to characterize aerosol compositions, water contents,

molecular interactions, and particle phases especially for the efflorescence process (Ma and He,

2012; Laskina et al., 2013; Zhou et al., 2014; Wang et al., 2015). In this study, the phase

transformations and hygroscopic properties of OA and mixed OA/AS droplets with varying OA

content were studied by confocal Raman spectroscopy in conjunction with optical microscopy.

Furthermore, we explored the effects of reactions between OA and AS on the chemical

compositions and hygroscopic properties of mixed OA/AS droplets.

**2 Experimental section**

**2.1 Sample preparation**

Ammonium sulfate (AS) and oxalic acid dihydrate were purchased from Sinopharm Chemical

Reagent Co. Ltd. (99.0% purity) and used without further purification. The 0.5 mol $L^{-1}$ pure

component AS and OA solutions were prepared by dissolving AS and oxalic acid dihydrate in

ultrapure water (18.2 MΩ·cm, Barnstead Easypure II), respectively. The mixed OA/AS solutions

with different organic to inorganic molar ratios (OIRs) of 1:3, 1:1 and 3:1 were obtained by

dissolving a designated amount of OA into AS solutions. The sample solution was discharged from

a syringe. Then, residual solution in the syringe was pushed rapidly to generate aerosol droplets

spraying onto a polytetrafluorethylene (PTFE) substrate fixed to the bottom of the sample cell. Then,

the sample cell was promptly sealed by a transparent polyethylene film. The RH in the sample cell

was regulated by nitrogen streams consisting of a mixture of water-saturated $N_2$ and dry $N_2$ at

controlled flow rates. At ~ 95% RH, the droplets with a diameter of 30 ~ 40 microns detected by an

optical microscope (50× objective, 0.75 numerical aperture) were selected to acquire the Raman spectra. The dry size of these particles after efflorescence ranged from 10 to 20 μm. The RH and temperature of the outflow from the sample cell was measured by a humidity/temperature meter (Centertek Center 313) with an accuracy of ±2.5% below 90% RH and ±0.7 K placed near the exit of the sample cell. The temperature accuracy of 0.7 K could result in uncertainty of 4% at RH of 95%. The temperature of the sample was maintained at 297 ± 0.5 K by using an automatic thermostat.

**2.2 Apparatus and conditions for the measurements**

The experimental setup used in this study was described in detail in previous work (Wang et al., 2008; Dong et al., 2009; Zhou et al., 2014). Briefly, the Renishaw InVia confocal Raman spectrometer equipped with a Leica DMLM microscope was used to acquire the Raman spectra. An argon-ion laser (wavelength 514.5 nm, model Stellar-REN, Modu-Laser) was used as an excitation source with an output power of 20 mW, and a 514.5 nm notch filter was adopted to remove the strong Rayleigh scattering. An 1800 g mm$^{-1}$ grating was used to obtain the spectra in the range of 200-4000 cm$^{-1}$ with a resolution of about 1 cm$^{-1}$. Spectral calibration was made using the 520 ±0.05 cm$^{-1}$ Stokes shift of silicon band before performing measurements. Then, spectroscopic measurements were made on droplets observed by using the Leica DMLM microscope with a 50× objective lens (0.75 numerical aperture). The spectra were obtained with three spectral scans, and each time with an accumulation time of 10 s. The sample droplets were injected onto the substrate at high RH (~ 95% RH). Subsequently, the RH was decreased stepwise for a slow dehydration process, and then increased stepwise from RH< 3% to high RH for a hydration process. The decrease rate was typically 5-6 RH/40 min, and the rate remained 2-3 RH/40 min near the phase transition. The RH was decreased continuously in a few minutes for a rapid dehydration process. The particles were equilibrated with water vapor at a given RH for about 40 min, during which the intensity ratios of the water peak (3430 cm$^{-1}$) to the sulfate peak (980 cm$^{-1}$) remained constant. The spectra of AS, OA and mixed OA/AS droplets were monitored and measured through a full humidity cycle. Multiple particles (three or four) were selected to acquire the Raman spectra through each humidity cycle. Each humidity cycle experiment was repeated at least three times. All the measurements were taken at ambient temperature of about 297 K.

Raman growth factor (g(RH)) is defined as the ratio of integrated area of OH stretching mode of

1  water (3350−3700 cm$^{-1}$) at each RH ($A_{RH}$) normalized to that of a dry particle ($A_{RH0}$) according to

2  Eq. (1) (Laskina et al., 2015).

$$g(\text{RH}) \; = \; A_{RH}/A_{RH0} \tag{1}$$

where $A_{RH}$ is integrated area of OH stretching mode from water (3350-3700 cm$^{-1}$) at a specific RH

and $A_{RH0}$ is that of a dry particle. Hygroscopic growth curves are acquired by plotting the average

Raman growth factor of duplicate particles as a function of RH.

**3 Results and discussion**

**3.1 Raman spectra of pure AS and OA droplets**

The Raman spectra of AS droplets during the dehydration and hydration process can be found in Fig.

1a and 1b, respectively. AS droplets effloresce at 44.3 ±2.5% RH, as indicated by the disappearance

of the water peak centered at 3437 cm$^{-1}$ and a red-shift in $\nu_s(SO_4^{2-})$ peak position from 979 to 974

12  cm$^{-1}$ during the dehydration process. For the hydration process, the deliquescence of AS particles is

observed to occur at 80.1 ±1.5% RH, resulting in an abrupt increase in the absorbance of the water

peak centered at 3437 cm$^{-1}$ and a blue-shift in $\nu_s(SO_4^{2-})$ peak position from 974 to 979 cm$^{-1}$.

The Raman spectra of OA droplets with varying RH during the dehydration and hydration

process are shown in Fig. 2, and the assignments of the peaks for OA are presented in Table 1

according to previous studies (Hibben, 1935; Ebisuzaki and Angel, 1981; Mohaček-Grošev et al.,

2009). As seen in Fig. 2a, the feature bands for OA droplets are observed at 1460, 1750 and 3433

19  cm$^{-1}$ at 92.5% RH. At lower RH around 71% (Fig. 2a, magenta line), these bands shift to 1490,

1737, 3433 and 3474 cm$^{-1}$, and a new band at 1689 cm$^{-1}$ occurs, which is entirely consistent with

the spectrum of oxalic acid dihydrate (Fig. 2a, black dashed line). It indicates OA droplets

crystallize to form oxalic acid dihydrate. Oxalic acid particles after efflorescence exist in the form

of dihydrate until 6.6% RH, at which the Raman spectrum of dihydrate remains unchanged for 40

24  min. Once RH decreases to ∼5.0%, the peaks promptly shift to 1477, 1710, 2587, 2760 and 2909

25  cm$^{-1}$, and peaks at 3433 and 3474 cm$^{-1}$ assigned to $\nu$(OH) vanish, which is the spectral feature of

anhydrous oxalic acid. This result implies that oxalic acid dihydrate is converted to anhydrous

oxalic acid in the RH around 5.0%. The Raman spectra of anhydrous oxalic acid particles during the

hydration process as a function of RH are shown in Fig. 2b. It can be found that the Raman spectra

feature for anhydrous oxalic acid particles occurs at RH<19.6%. At 19.6% RH, the peaks observed

at 1490, 1737, 3433 and 3474 cm$^{-1}$ are identical to that of oxalic acid dihydrate (Fig. 2a, black

dashed line), indicating the formation of oxalic acid dihydrate. The observation of no spectral change until 94% RH suggests that oxalic acid dihydrate shows no deliquescence transition in the 0-94 % RH range studied, consistent with previous studies (Peng et al., 2001; Braban et al., 2003; Ma et al., 2013a; Jing et al., 2016). The transition point of anhydrous oxalic acid to oxalic acid dihydrate upon hydration is 17.9-19.6% (Fig. 2b), in agreement with the results reported by Braban et al. (2003) and Ma et al. (2013a).

**3.2 Raman spectra of OA/AS mixtures**

The Raman spectra of mixed OA/AS droplets with OIRs of 1:3, 1:1 and 3:1 at various RHs during the dehydration and hydration process are depicted in Fig. 3 and 4, respectively. Since spectral features upon hydration are identical to the dehydration process, here we only analysed spectral evolution of efflorescence process in detail. The detailed assignments are summarized in Table 2. For the mixed OA/AS droplets (OIR = 1:3) at 96.2% RH (seen in Fig. 3a), the bands at 979 cm$^{-1}$ and 1049 cm$^{-1}$ are characteristic peaks of aqueous $SO_4^{2-}$ and $HSO_4^-$ ($v_s(SO_3)$), respectively. In addition, the peak at 1741 cm$^{-1}$ and 1446 cm$^{-1}$ can be assigned to vibrating mode of aqueous OA and $HC_2O_4^-$, respectively. With decreasing RH, only small changes are observed in the spectra until the RH reaches 34.4% RH. At 34.4% RH, the shift of $v_s(SO_4^{2-})$ peak from 979 cm$^{-1}$ to 974 cm$^{-1}$ indicates the crystallization of AS, as also seen in Fig. 10b. A new band centered at 874 cm$^{-1}$ corresponds to combination bands of the vibrational mode ($\delta(S-OH)$) of $HSO_4^-$ ion from $NH_4HSO_4$ (Dawson et al., 1986) and $HC_2O_4^-$ ion vibrating (Villepin and Novak, 1971), suggesting the formation of crystalline $NH_4HC_2O_4$. Moreover, the several new peaks at 1416, 1469 and 1660 cm$^{-1}$ can be attributed to the $HC_2O_4^-$ ion vibrating of crystalline $NH_4HC_2O_4$ (Villepin and Novak, 1971). Therefore, the evolution of Raman spectra of the mixed OA/AS droplets (OIR = 1:3) during the dehydration process confirms that OA could react with AS to form $NH_4HSO_4$ and $NH_4HC_2O_4$, which supports previous speculation for the reaction between OA and AS (Miñambres et al., 2013). The reaction of OA with AS occurs via the following pathway:

$$(NH_4)_2SO_4(aq) + H_2C_2O_4(aq) \rightarrow NH_4HSO_4(aq) + NH_4HC_2O_4(aq)$$

For the mixed OA/AS droplets (OIR = 1:1, Fig. 3b), the evolution of spectra shows resemblance to that of mixed droplets (OIR = 1:3). At 96.1% RH, the peaks at 979 cm$^{-1}$, 1751 cm$^{-1}$, 1051 cm$^{-1}$ and 1448 cm$^{-1}$ can be assigned to vibrating mode of $SO_4^{2-}$, OA, $HSO_4^-$ ($v_s(SO_3)$) and $HC_2O_4^-$, respectively. At 75.0% RH, a new peak at 874 cm$^{-1}$ corresponding to the vibrational mode ($\delta(S-OH)$)

of $HSO_4^-$ and the $HC_2O_4^-$ ion vibrating as well as the new peaks at 494, 1469 and 1677 cm$^{-1}$ due to the $HC_2O_4^-$ vibrating mode, indicates that crystalline $NH_4HC_2O_4$ is generated from the reaction of OA with AS. As the RH further decreases to 44.3%, the $v_s(SO_4^{2-})$ band shifts from 979 cm$^{-1}$ to 974 cm$^{-1}$, indicating the formation of crystallized AS particles.

For the mixed OA/AS droplets (OIR = 3:1, Fig. 3c) at 95.9% RH, the bands at 980 cm$^{-1}$, 1752 cm$^{-1}$ and 1050 cm$^{-1}$ are characteristic peak of the $SO_4^{2-}$ ion, OA and $HSO_4^-$ ion ($v_s(SO_3)$), respectively. And the peak at 1460 cm$^{-1}$ can be attributed to vibrating mode of $HC_2O_4^-$ ion. When the RH decreases to 74.4%, a new band at 874 cm$^{-1}$ is contributed by the vibrational mode of both $HSO_4^-$ ($\delta$(S-OH)) and $HC_2O_4^-$. Meanwhile, the bands at 494, 1471 and 1654 cm$^{-1}$ can be attributed to $HC_2O_4^-$ vibrating mode, suggesting OA reacts with AS to yield crystalline $NH_4HC_2O_4$ during the dehydration process. At 64.4% RH, the peaks at 494, 874, 1471, 1654, 1718 cm$^{-1}$, and the peak at 3426 cm$^{-1}$ from oxalic acid dihydrate become sharp and narrow, indicating that the OA/AS droplets (OIR = 3:1) completely crystallize to form $NH_4HC_2O_4$ and $H_2C_2O_4$ 2$H_2O$. No obvious change in spectral feature of the major bands is observed with RH decreasing from 64.4% to 1.1%.

**3.3 Hygroscopicity of pure AS, OA and OA/AS mixtures**

**3.3.1 Phase transitions and chemical transformation of AS in mixed systems**

Considering that the peak position is sensitive to the chemical environment in the aerosols, the position of the $v_s(SO_4^{2-})$ mode can be used to determine the phase transitions of AS. The previous studies have also applied the abrupt shift in characteristic peak position to indicate phase transition of ammonium sulfate during the hygroscopic process (Braban and Abbatt, 2004; Ling and Chan, 2008; Yeung et al., 2009). Figure 5 presents the peak position of the $v_s(SO_4^{2-})$ for AS droplets and mixed OA/AS droplets during the dehydration and hydration process, respectively. During the dehydration process, a red shift from 979 to 974 cm$^{-1}$ can be observed for AS and OA/AS mixed particles with OIRs of 1:3 and 1:1, indicating crystallization of AS from droplets. During the hydration process, the observations of blue shift from 974 to 979 cm$^{-1}$ for AS and OA/AS mixed particles with OIRs of 1:3 and 1:1 suggest the deliquescence transition of AS from crystal phase to aqueous solution. For OA/AS mixed particles with an OIR of 3:1, the peak shift between ~966 and ~979 cm$^{-1}$ is determined during the whole RH cycle. The shift of $v_s(SO_4^{2-})$ mode to 966 cm$^{-1}$ suggests the formation of letovicite $(NH_4)_3H(SO_4)_2(s)$ (Damak et al., 1985). The DRH and ERH for pure and mixed systems have been shown in Fig. 5 and detailed discussion is given in the following

section.

The peaks at ~1049 and ~979 cm$^{-1}$ for mixed OA/AS droplets (OIRs = 1:3, 1:1 and 3:1) can be

attributed to the $HSO_4^-$ and $SO_4^{2-}$ stretching mode, respectively. The area ratio of Raman peaks

assigned to the $HSO_4^-$ and $SO_4^{2-}$ is used to indicate the degree of conversion of $SO_4^{2-}$ into $HSO_4^-$

($\alpha_{HSO4-}$) in mixtures, which can be expressed as following:

$$\alpha_{HSO_4^-} = A_{1049} / (A_{1049} + A_{979}) \tag{2}$$

where $A_{1049}$ and $A_{979}$ is the peak area of the $HSO_4^-$ and $SO_4^{2-}$, respectively. The ~1049 cm$^{-1}$ for

$HSO_4^-$ at solid mixture is not obvious compared to that for solutions. Thus, the calculations are

based on the bands at RH approaching the full efflorescence point. The estimated $\alpha_{HSO4-}$ value for

OIR = 1:3 (36.1% RH), OIR = 1:1 (46.2% RH) and OIR = 3:1 (66.2% RH) is 0.048, 0.368 and

0.644, respectively, indicating the enhanced conversion of $SO_4^{2-}$ into $HSO_4^-$ with increasing OA

content in the mixed systems. Due to the effects of Raman cross section, $\alpha_{HSO4-}$ could not represent

the actual degree of conversion. In fact, here $\alpha_{HSO4-}$ is only used for comparisons of degree of

conversion of $SO_4^{2-}$ into $HSO_4^-$ between mixed particles with varying OIRs.

**3.3.2 Hygroscopic growth of pure and mixed components**

Hygroscopicity curves of AS and OA particles are shown in Fig. 6. The optical images of the AS

particle at the phase change points can be seen in Fig. 7. The ERH of AS is determined to be 44.3 ±

2.5% RH, which generally falls into the range from 33 to 52% RH reported in the literature (Tang

and Munkelwitz, 1994a; Cziczo et al., 1997; Dougle et al., 1998; Laskina et al., 2015). The DRH of

AS particles is observed to occur at 80.1 ± 1.5% RH, which agrees well with reported values of

80% RH by EDB (Tang and Munkelwitz, 1994a) and 82.3 ± 2.5% RH by micro-Raman

spectroscopy (Laskina et al., 2015). As shown in Fig. 6b and Fig. 8, the measured ERH of OA is 71

± 2.5% RH, which deviates much from the reported value of 51.8-56.7% RH by Peng et al. (2001)

using the EDB technology. It is worthwhile to point out that the conversion of OA droplets to oxalic

acid dihydrate at 71% RH is inconsistent with the observation of Peng et al. (2001). They observed

that OA droplets crystallized to form anhydrous oxalic acid rather than oxalic acid dihydrate at

51.8-56.7% RH. The discrepancy on the ERH of OA compared to that reported by Peng et al. (2001)

is likely due to the effects of substrate and sample purity. The size of dry particles ranging from 10

to 20 μm in our experiment is consistent with observation using EDB by Peng et al. (2001), which

eliminates the influence of particle size. The substrate supporting droplets may promote the

heterogeneous nucleation of oxalic acid while the levitated droplets in EDB study can avoid induced nucleation by the substrate. Ghorai et al. (2014) also reported the potential effects of substrate on the efflorescence transition of NaCl/dicarboxylic acid mixed particles. In addition, The OA purity in our study is 99.0% lower than that of 99.5% in study by Peng et al. (2001). Thus, trace amounts of impurities in OA droplets acting as a heterogeneous nucleus could contribute to crystallization and result in a higher ERH of OA. Due to the effects of substrate and sample purity, the heterogeneous nucleation should be responsible for the discrepancy on the observed ERH of OA. The water content of the supersaturated droplet at the onset of crystallization determines the form of oxalic acid crystal generated, i. e., anhydrous OA or OA dihydrate. Due to a higher ERH, oxalic acid droplets with more water content favor the formation of a dihydrate after crystallization. It should be noted that our experiment appears to be favored in the atmospheric environment, considering that insoluble material such as mineral dust mixed with OA may play the role of substrate thus facilitating the heterogeneous nucleation of OA aerosols. The Raman growth factor of OA shows no obvious change between ~71% and 6.6% RH upon dehydration. At RH lower than 5%, the Raman growth factors drop abruptly due to the transformation of crystalline $H_2C_2O_4 \cdot 2H_2O$ into anhydrous oxalic acid, as also indicated by Raman spectrum. It seems that the structure of anhydrous OA particle is not as compact as that of dihydrate, seen in Fig. 8. Thus, the loss of crystal water results in no obvious change in particle size. During the hydration process, the Raman growth factor of OA shows a slightly increase at 19.6% RH, which can be attributed to the conversion of anhydrous oxalic acid to dihydrate. The transition point of anhydrous oxalic acid to oxalic acid dihydrate agrees with previous studies (Braban et al., 2003; Ma et al., 2013b; Miñambres et al., 2013). No deliquescence behavior is observed for oxalic acid dihydrate even at 94% RH, consistent with early observations (Ma et al., 2013b; Miñambres et al., 2013; Jing et al., 2016).

Figure 9 presents hygroscopic growth of OA/AS mixtures with OIRs of 1:3, 1:1 and 3:1. As can be seen in Fig. 9a and 10b, mixed OA/AS droplets (OIR = 1:3) exhibit efflorescence transition at lower $34.4 \pm 2.0\%$ RH relative to ERH ($44.3 \pm 2.5\%$) of pure AS. During the hydration process, mixed particles start to absorb slight water before deliquescence at $81.1 \pm 1.5\%$ RH (seen in Fig. 9 and 10). It can be seen in Fig. 10 that the size of 1:3 mixed OA/AS particle at 79.4% RH prior to deliquescence appears to be larger than that after complete efflorescence. The decrease in ERH and slight water uptake before deliquescence for 1:3 mixed particles is likely due to the effects of

NH₄HSO₄ formed upon dehydration. $NH_4HSO_4$ has a low ERH (22-0.05%) and DRH (40%) (Tang and Munkelwitz, 1994a), which may affect the nucleation and crystallization of AS upon dehydration and lead to slight water uptake prior to the deliquescence of AS. The hygroscopic growth of mixed particles upon dehydration is in fair agreement with that of pure AS or OA. However, the Raman growth factors of mixed particles upon hydration show a considerable decrease in comparison to that upon dehydration. The discrepancies for Raman growth factor at high RH between the two processes can be attributed to the formation of $NH_4HC_2O_4$ and residual solid OA, both of which have a high deliquescence point larger than 95% RH (Schroeder and Beyer, 2016). During the hydration process, $NH_4HC_2O_4$ and OA in the mixed aerosols remains solid even at high RH (also seen in Fig. 10d), resulting in less water uptake of mixed particles. The similar phenomenon is also observed for NaCl/OA mixed particles upon hydration due to the formation of less hygroscopic sodium oxalate (Peng et al., 2016).

The mixed OA/AS droplets with an OIR = 1:1 first partially effloresce at 75.0% ±1.6% due to the crystallization of $NH_4HC_2O_4$, as indicated by Raman spectra. Then, the full efflorescence occurs at 44.3 ±2.5% RH with the crystallization of AS. The full ERH of 1:1 OA/AS mixed droplets is highly consistent with that of pure AS. During the hydration process, the Raman growth factor of 1:1 mixed particles increases slightly at 35.5% RH, and then remains almost invariable until 77% RH, which is likely due to the formation of hydrate. The deliquescence transition occurs at 77 ± 1.0% RH slightly lower than DRH of AS, which agrees with literature results for AS particles containing OA (Brooks et al., 2002; Jing et al., 2016). The water contents of mixed droplets after deliquescence are significantly lower than those upon dehydration. The Raman features at 494 cm⁻¹ and 874 cm⁻¹ have confirmed the presence of solid $NH_4HC_2O_4$ upon hydration across all RHs studied (seen in Fig. 4), which should be responsible for the decreasing water uptake of the mixed particles at high RH.

For mixed OA/AS droplets with an OIR = 3:1, the partial and full efflorescence transition could be observed at 74.4 ±1.0% RH and 64.4 ±3.0% RH, respectively (seen in Fig. 9 and 11). As seen in Fig. 3c, the bands at 494, 1471 and 1654 cm⁻¹ suggest the formation of crystalline $NH_4HC_2O_4$ at 74.4 ±1.0% RH. Figure 12 presents the spatial distribution of chemicals within mixed OA/AS (OIR = 3:1) particles at 74.4% RH. The characteristic peak of 980 cm⁻¹, 1050 cm⁻¹ and 1471 cm⁻¹ is assigned to $SO_4^{2-}$, $HSO_4^-$ and $HC_2O_4^-$, respectively. The sharp absorption at 874 cm⁻¹ and obvious

peak at 1471 cm$^{-1}$ indicate the abundant content of $NH_4HC_2O_4$. The comparison of characteristic peaks between inner and outer phase reveals that the major component on the surface of a mixed OA/AS (OIR = 3:1) particle is $NH_4HC_2O_4$. In contrast to the surface, the obvious features of 980 cm$^{-1}$ and 1050 cm$^{-1}$ at the core of the particle suggest that $(NH_4)_2SO_4$ and $NH_4HSO_4$ mainly exist in the inner aqueous phase. During the dehydration process, crystalline $NH_4HC_2O_4$ in the outer phase acts as the heterogeneous nucleus, leading to the crystallization of oxalic acid dihydrate and other components in the inner phase. Thus, the full ERH of 3:1 OA/AS mixed droplets is higher than that of pure AS (44.3 ± 2.5% RH) and $NH_4HSO_4$ (22-0.05% RH). During the hydration process, Raman growth factors of mixed particles slightly increase at 34.5% RH. No deliquescence transition or significant water uptake is observed over the RH range studied. This phenomenon can be explained by the fact that the most of AS in the mixtures has been converted into $NH_4HC_2O_4$ and $NH_4HSO_4$ or letovicite. Although $NH_4HSO_4$ with a low DRH may contribute to water uptake of mixed particles, the minor $NH_4HSO_4$ or letovicite formed in the mixtures is likely to be coated by $NH_4HC_2O_4$ and OA with a high DRH. Thus, the mixed OA/AS particles with an OIR = 3:1 show no obvious hygroscopic growth upon hydration due to the change in aerosol composition and morphology effects. The effects of morphology on the hygroscopic growth of aerosols have been reported for AS particles containing adipic acid (Sjogren et al., 2007). The water uptake of AS particles containing relatively high content of adipic acid could be suppressed due to AS enclosed by the crust of solid adipic acid with a high DRH.

The observed efflorescence relative humidity (ERH) for mixed droplets was dependent on the molar ratio of oxalic acid to ammonium sulfate. The mixed OA/AS droplets with an OIR of 1:3 are observed to effloresce completely at 34.4 ± 2.0% RH relative to ERH of pure AS (44.3 ± 2.5%) or OA (71 ± 2.5%). It can be seen that AS as a major fraction of the particle does not promote the heterogeneous nucleation of OA. Meanwhile, the crystallization of AS is also influenced due to the presence of OA. The similar phenomenon was also observed for malonic acid/ammonium sulfate mixtures with minor organic content (Braban and Abbatt, 2004; Parsons et al., 2004). Braban and Abbatt (2004) found that the ERH of malonic acid/ammonium sulfate mixed particles was considerably decreased compared to that of pure ammonium sulfate for mass fractions of malonic acid less than 0.3. They concluded that the presence of ammonium sulfate in the supersaturated droplet could exert the extra barrier to nucleation of malonic acid crystals rather than play the role

of a heterogeneous nucleation site. As for 1:3 OA/AS mixed droplets, ammonium sulfate may also inhibit the nucleation of oxalic acid at relatively high RH. With decreasing RH, aqueous oxalic acid could enhance the viscosity of the droplet due to hydrogen bond interactions (Mikhailov et al., 2009), thus limiting the nucleation of ammonium sulfate and resulting in a lower ERH with respect to the value of pure AS (Parsons et al., 2004). In the case of mixed OA/AS droplets with an OIR of 1:1 and 3:1, the $NH_4HC_2O_4$ formed at ~75% RH upon dehydration likely acts as a heterogeneous nucleus for crystallization of other components, which increases full efflorescence point of mixed particles. One study indicated that Aldrich humic acid sodium salt (NaHA) could also promote the ERH of ammonium sulfate (Badger et al., 2006). Similar to oxalic acid, succinic acid and adipic acid have a high deliquescence point and low solubility. However, it has been found that the efflorescence point of ammonium sulfate in mixed particles is not elevated even when the content of succinic acid or adipic acid is more than 50% by mass or mole fractions (Ling and Chan, 2008; Yeung et al., 2009; Laskina et al., 2015). The chemical nature of solid determines its ability to act as a heterogeneous nucleus (Braban and Abbatt, 2004). In contrast to ammonium sulfate particles containing succinic acid or adipic acid, our results suggest that the addition of oxalic acid into ammonium sulfate droplets may trigger partial and full crystallisation of aerosols at relatively higher RH upon dehydration due to $NH_4HC_2O_4$ product acting as an effective nucleus.

During the deliquescence process, the OA/AS mixed particles with an OIR of 1:3 and 1:1 exhibit a slightly lower deliquescence point than that of pure ammonium sulfate, consistent with previous observations of effects of crystalline oxalic acid on deliquescence transition of ammonium sulfate (Brooks et al., 2002; Wise et al., 2003; Jing et al., 2016). It should be noted that prior literature result also showed that continuous or smooth water uptake from low RH was observed for particles composed of AS and OA with a mass ratio of 1.5:1 due to the fact that after drying processing oxalic acid existing in an amorphous or liquid-like state prevented nucleation of ammonium sulfate even under dry conditions (Prenni et al., 2003). In the present study, water uptake by the OA/AS mixed particles at high RH upon hydration is dramatically lower than that upon dehydration and significantly decreased with elevated OA content. This phenomenon distinguishes from hygroscopic characteristics of typical water-soluble mixtures in literatures. It has been found that hydration growth curve and dehydration growth curve are typically merged above deliquescence point for mixed systems containing inorganic salts and water-soluble organic compounds (Choi and Chan,

2002; Chan and Chan, 2003; Gysel et al., 2004; Clegg and Seinfeld, 2006; Sjogren et al., 2007; Pope et al., 2010; Ghorai et al., 2014; Estillore et al., 2016). In this study, Raman spectra and the micrograph suggest the presence of solid $NH_4HC_2O_4$ and residual solid OA at high RH should be responsible for the decreased water uptake during the hydration process. In contrast, Prenni et al. (2003) reported that the hygroscopic growth of OA/AS mixed particles remained unchanged at 90% RH with OA mass fraction ranging from 0.01 to 0.4. In addition, they also found that water uptake after deliquescence was well described by the model method assuming complete dissolution of OA in aqueous phase as well as no interactions between OA and AS, which was also observed by Jing et al. (2016) using the HTDMA. The previous HTDMA studies for OA/AS mixed particles indicate no composition change and no specific interactions existing between OA and AS (Prenni et al., 2003; Jing et al., 2016). However, it should be noted that the HTDMA studies did not perform measurements for the dehydration process such that aerosols underwent rapid drying on the time scale of seconds, i.e., the total residence time for transformation of droplets into dry particles in the drying section of HTDMA is typically tens of seconds (Prenni et al., 2003; Jing et al., 2016), much shorter than that (10 ~ 12 h) in our study. In the HTDMA experiments, the combination of faster drying and smaller particles with submicron size implies that the aqueous phase obtained higher supersaturations than in our present study (Rosenoern et al., 2008), leading to less dissociation of oxalic acid and thus less $HC_2O_4^-$ formed in the droplets as well as the inhibited formation of $NH_4HC_2O_4$. The fast evaporation of water from the surface of an aqueous droplet upon rapid drying could result in a higher surface concentration of solutes than the slow drying process (Treuel et al., 2011). The higher surface concentration of oxalic acid corresponds to less formation and hence decreased supersaturation of $HC_2O_4^-$. Due to the dependence of nucleation rate on the extent of supersaturation, it can be expected that the nucleation of $NH_4HC_2O_4$ is suppressed within OA/AS mixed droplets undergoing rapid drying.

Considering the potential effects of drying time on the reactions between OA and AS, we explored the hygroscopicity of OA/AS particles with an OIR of 1:1 after rapid drying process. The mixed OA/AS droplets undergo dehydration to form dry particles in 3 ~ 5 min. We observed one-step efflorescence of rapidly-dried particles (1:1, molar ratio) occurred at 47% $\pm$ 2.5% RH, compared to the two-step efflorescence of slowly-dried particles occurring at 75% and 44.3% RH, respectively. The Raman spectra and hygroscopic curve upon hydration for OA/AS particles with an

OIR of 1:1 are presented in Fig. 13. The obvious discrepancies can be observed for spectra at ~2% RH between the two drying processes. After rapid drying process, the spectra at ~2% RH show the feature of crystalline AS (974 $cm^{-1}$, $\nu_s(SO_4^{2-})$)) and anhydrous OA (1710 $cm^{-1}$, $\nu(C=O)$; 1479 $cm^{-1}$, $\nu_s(COO)$). Meanwhile, no characteristic peaks for $NH_4HC_2O_4$ (494 $cm^{-1}$, $\delta(COO)$; 874 $cm^{-1}$, $\nu(C-C)$; 1729 $cm^{-1}$, $\nu(C=O)$; 1469 $cm^{-1}$, $\nu_s(COO)$) and $NH_4HSO_4$ (874 $cm^{-1}$, $\delta(S-OH)$) can be identified in the spectra. It is clear that the drying time for transformation of droplets into dry particles has impacts on the reactions of OA with AS in the aerosols due to particle-phase processes under kinetic control. Previous studies found the longer drying time could lead to greater nitrate depletion between nitrates and organic acids, which results from slow reaction and diffusion in the viscous aerosols (Wang and Laskin, 2014). The Raman growth factors of mixed particles with an OIR of 1:1 also increase slightly at 36.5% RH due to the formation of OA dihydrate, as indicated by the Raman feature. The deliquescence transition of mixed particles occurs at 79.3% RH. After deliquescence, Raman growth factors of mixed particles after rapid drying process are lower than that after slow drying process, which may be caused by the fact that at high RH the hygroscopic growth of AS is slightly lower than that of $NH_4HSO_4$ formed in the particles after slow drying process (Tang and Munkelwitz, 1977). In addition, it is found that after deliquescence OA dihydrate remains solid in the mixed particles undergoing rapid drying.

**4 Conclusions and atmospheric implications**

In this work, confocal Raman spectroscopy is used to investigate the hygroscopic properties and phase transformations of OA and internally mixed OA/AS droplets (OIRs = 1:3, 1:1 and 3:1). OA droplets effloresce to form oxalic acid dihydrate at 71 ±2.5% RH, and then oxalic acid dihydrate further loses crystalline water to form anhydrous oxalic acid at ~5.0% RH during the dehydration process. The Raman spectra of mixed OA/AS droplets reveal the formation of $NH_4HC_2O_4$ and $NH_4HSO_4$ from the reaction of OA with AS in aerosols during the dehydration process. The deliquescence and efflorescence point for AS is observed to occur at 80.1 ±1.5% and 44.3 ±2.5% RH, respectively. The ERH of the mixed OA/AS droplets with 1:3, 1:1 and 3:1 ratio is determined to be 34.4 ±2.0%, 44.3 ±2.5% and 64.4 ±3.0% RH, respectively, indicating significant effects of OA content on the efflorescence transition of AS. The mixed OA/AS particles with 1:3 and 1:1 ratio show deliquescence transition at 81.1 ±1.5% and 77 ±1.0% RH, respectively, which is close to the DRH of AS. The mixed OA/AS particles with 3:1 ratio exhibit no deliquescence transition over the

RH range studied due to the transformation of $(NH_4)_2SO_4$ into high-DRH $NH_4HC_2O_4$. The hygroscopic growth of mixed particles at high RH upon hydration is substantially lower than that of corresponding dehydration process and further decreases with increasing OA content. The discrepancies for water content of mixed particles between the two processes at high RH can be explained by the significant formation of low hygroscopic $NH_4HC_2O_4$ and residual OA, which still remain solid and thus result in less water uptake of mixed particles.

The prior hygroscopic studies suggest that crystallization of internally mixed ammonium sulfate/dicarboxylic acid particles may lead to the formation of trace organic salt. Lightstone et al. (2000) estimated that approximately 2% of the initial succinic acid may form ammoniated succinate within mixed ammonium nitrate/succinic acid particles during the efflorescence process. Ling and Chan (2008) inferred that crystallization of ammonium sulfate/succinic acid droplets likely generated metastable organic salt based on change in the Raman peak form of succinic acid. Braban and Abbatt (2004) reported that $NH_4HSO_4$ and ammoniated malonate were likely generated upon crystallization of mixed ammonium sulfate/malonic acid particles. However, due to the trace amount of organic salt below Raman or infrared detection limit, they found no apparent influence of organic salt formed upon dehydration on the water uptake or phase change of mixed particles. In contrast, our results indicate that the chemical processing upon drying of droplets containing OA and AS influences efflorescence transition and water uptake of mixed aerosols during the humidity cycle by modifying particulate component.

Our results highlight the atmospheric importance of dicarboxylic acid−ammonium sulfate interactions in aerosol aqueous chemistry. Such chemical processing upon drying of aerosols comprised of organic acid/$(NH_4)_2SO_4$ mixtures may enhance the acidity of aqueous phase in the intermediate RH due to the transformation of $(NH_4)_2SO_4$ into $NH_4HSO_4$. These experiments also imply that the chemical reaction between aqueous $(NH_4)_2SO_4$ and oxalic acid upon slow dehydration is a possible formation pathway for the low-volatility oxalate in ambient particles, which could enhance partitioning of dicarboxylic acids to aqueous particles with the presence of ammonium sulfate (Yli-Juuti et al., 2013; Hakkinen et al., 2014). It has been reported that the aerosol aqueous processing within organic acid/AS mixtures partly contributes to enhanced loadings of secondary organic aerosol (SOA) from biogenic precursors (Hoyle et al., 2011). Compared to aqueous processing such as condensed phase acid-catalyzed reactions relevant to

formation of organosulfates, the contribution of other aerosol processing containing organic salt formation to SOA burden likely becomes important under less acidic condition. Formation of low-solubility organic salts from aqueous processing within aerosols alters particle-phase component and thus modifies aerosol's hygroscopicity, optical properties and chemical reactivity. Our findings provide fundamental insight into effects of drying conditions (drying rate or time) on formation of organic salt from reactions of organic acids with inorganic salts in particle phase under ambient RH conditions. Overall, a better understanding of the chemical interactions between species in a multicomponent system during the humidity cycle is critical for the accurate modeling efforts of aerosol phase behavior in thermodynamic models.

*Data availability.* All data are available upon request from the corresponding authors.

*Competing interests.* The authors declare that they have no conflict of interest.

*Author contribution. YZ, MG and BJ designed the experiments and XW carried them out. XW and BJ performed the data analysis and prepared the manuscript with contributions from all co-authors.*

*Acknowledgments.* This project was supported by the National Natural Science Foundation of China (Contract No. 91544223, 21473009, and 21373026) and the National Key Research and Development Program of China (2016YFC0202202).

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

| Solid $H_2C_2O_4$ | | $H_2C_2O_4$ | $(NH_4)_2SO_4$ | | |
|---|---|---|---|---|---|
| Anhydrous | Dihydrate | Droplets (92.5% RH) | Droplets (94.8% RH) | Refs | Assignments |
| | | | 450 | (Spinner, 2003) | $\delta_s(SO_4^{2-})$ |
| 482 | 477 | 457 | | (Hibben, 1935) | $\delta(OCO)$ |
| 828 | | | | (Ebisuzaki and Angel, 1981) | $r(OCO)$ |
| 845 | 855 | 845 | | (Ebisuzaki and Angel, 1981) | $v(C\text{-}C)$ |
| | | | 979 | (Spinner, 2003) | $v_s(SO_4^{2-})$ |
| 1477 | 1490 | 1460 | | (Ebisuzaki and Angel, 1981) | $v_s(COO)$ |
| | 1627 | 1636 | | (Ebisuzaki and Angel, 1981) | $\delta(HOH)$ |
| | 1689 | | | (Ebisuzaki and Angel, 1981) | $v(C=O)$ |
| 1710 | 1737 | 1750 | | (Hibben, 1935) | $v(C=O)$ |
| 2587, 2760 | | | | (Mohaček-Grošev et al., 2009) | Combinations |
| 2909 | | | | | |
| | | | 3080 | (Spinner, 2003) | Combinations |
| | | | 3240 | (Spinner, 2003) | $v(OH)$ |
| | 3433, 3474 | 3433 | 3437 | (Spinner, 2003; Ebisuzaki and Angel, 1981) | $v(OH)$ |

v: stretching; δ: bending; r: rocking; s: symmetric mode.

**Table 2.** Molecular vibration assignments of mixed oxalic acid/ammonium sulfate systems

| $H_2C_2O_4$-$(NH_4)_2SO_4$ (1:3), RH = 96.2% | $H_2C_2O_4$-$(NH_4)_2SO_4$ (1:1), RH = 96.1% | $H_2C_2O_4$-$(NH_4)_2SO_4$ (3:1), RH = 95.9% | Refs | Assignments |
|---|---|---|---|---|
| 450 | 450 | 461 | (Spinner, 2003) | $\delta_s(SO_4^{2-})$ |
|  | 852 | 850 | (Ebisuzaki and Angel, 1981) | $v(C\text{-}C)$ |
| 979 | 979 | 980 | (Spinner, 2003) | $v_s(SO_4^{2-})$ |
| 1049 | 1051 | 1050 | (Dawson et al., 1986) | $v_s(SO_3)$ |
|  | 1382 | 1382 | (Chang and Huang, 1997) | $\omega(OCO)$ |
| 1446 | 1448 | 1460 | (Ebisuzaki and Angel, 1981) | $v_s(COO)$ |
| 1694 |  |  | (Ebisuzaki and Angel, 1981) | $v(C\text{=}O)$ |
| 1741 | 1751 | 1752 | (Ebisuzaki and Angel, 1981) | $v(C\text{=}O)$ |
| 3430 | 3427 | 3426 | (Spinner, 2003) | $v(OH)$ |

$v$: stretching; $\delta$: bending; $\omega$: wagging; s: symmetric mode.

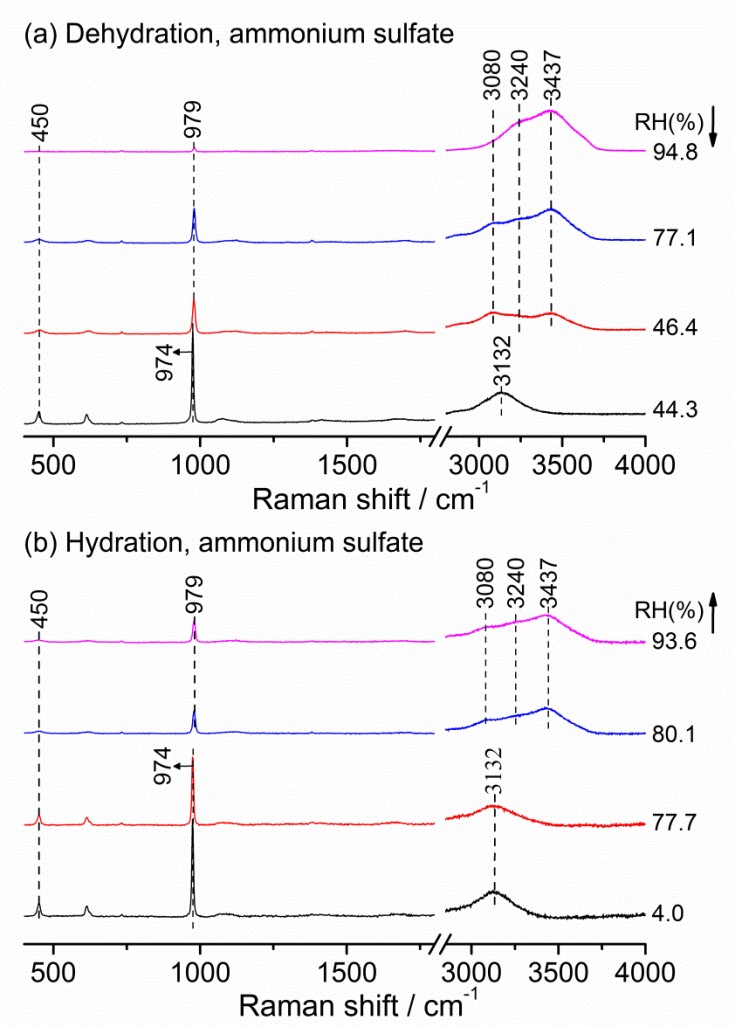

**Figure 1.** Raman spectra of ammonium sulfate droplets at various RH values during the (a) dehydration process and (b) hydration process.

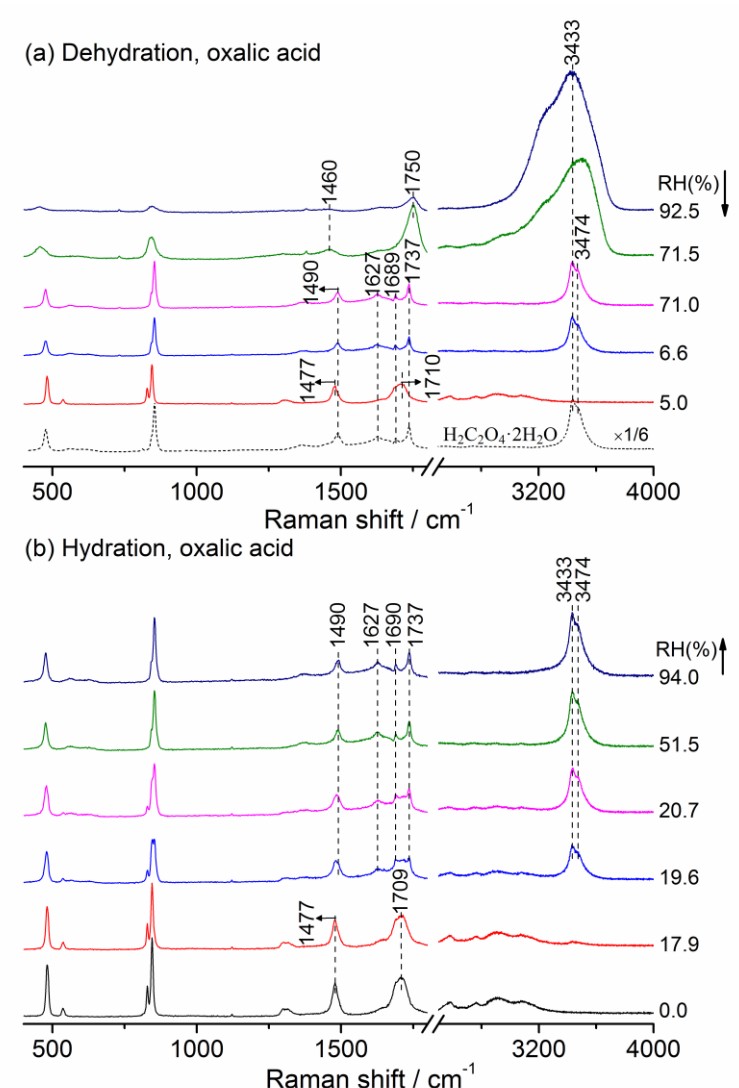

**Figure 2.** Raman spectra of oxalic acid droplets during the (a) dehydration process and (b) hydration process. In panel (a), the black dashed line indicates the spectrum of pure $H_2C_2O_4 \cdot 2H_2O$ particles with the peak height of $\nu(OH)$ located at 3433 cm$^{-1}$ scaled by a factor of 1/6.

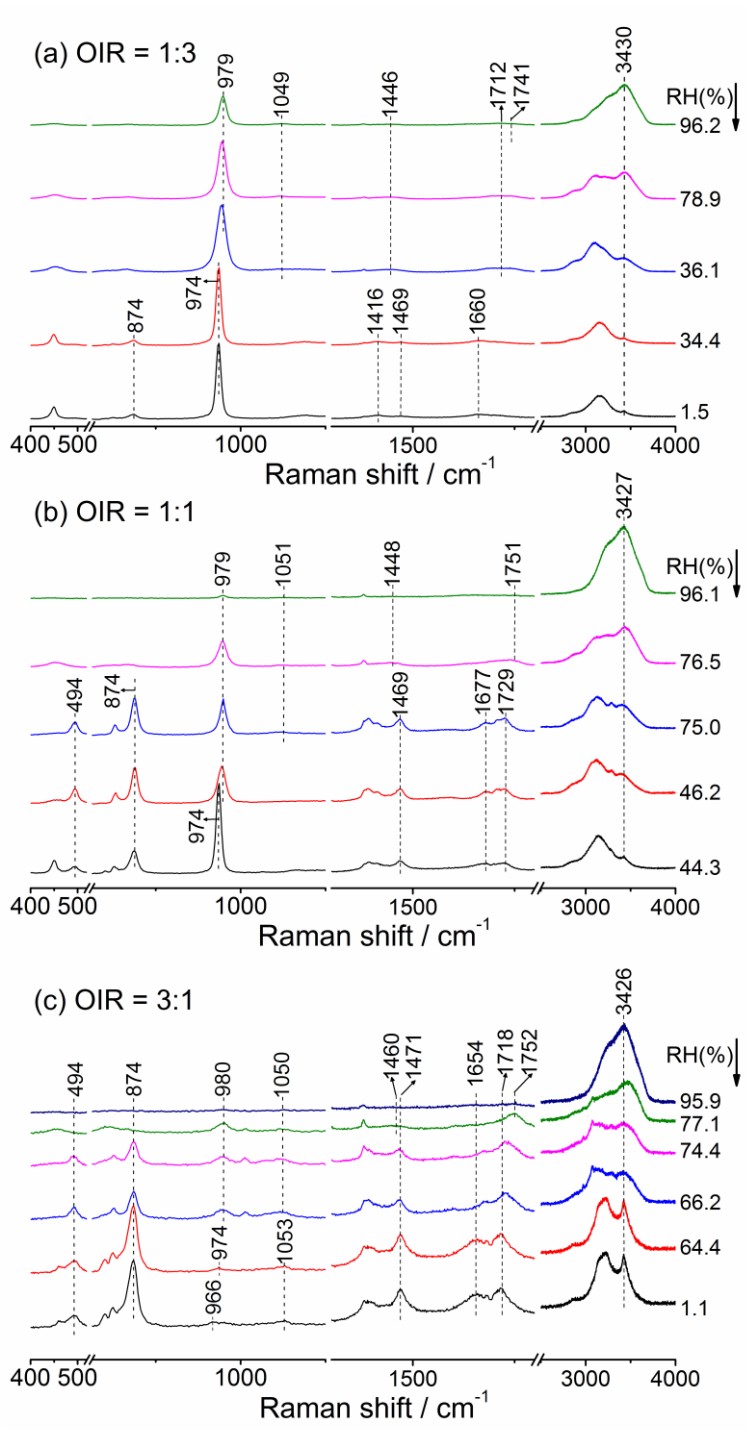

**Figure 3.** Raman spectra of mixed oxalic acid/ammonium sulfate droplets with OIRs of (a) 1:3, (b) 1:1 and (c) 3:1 at various RH values during the dehydration process.

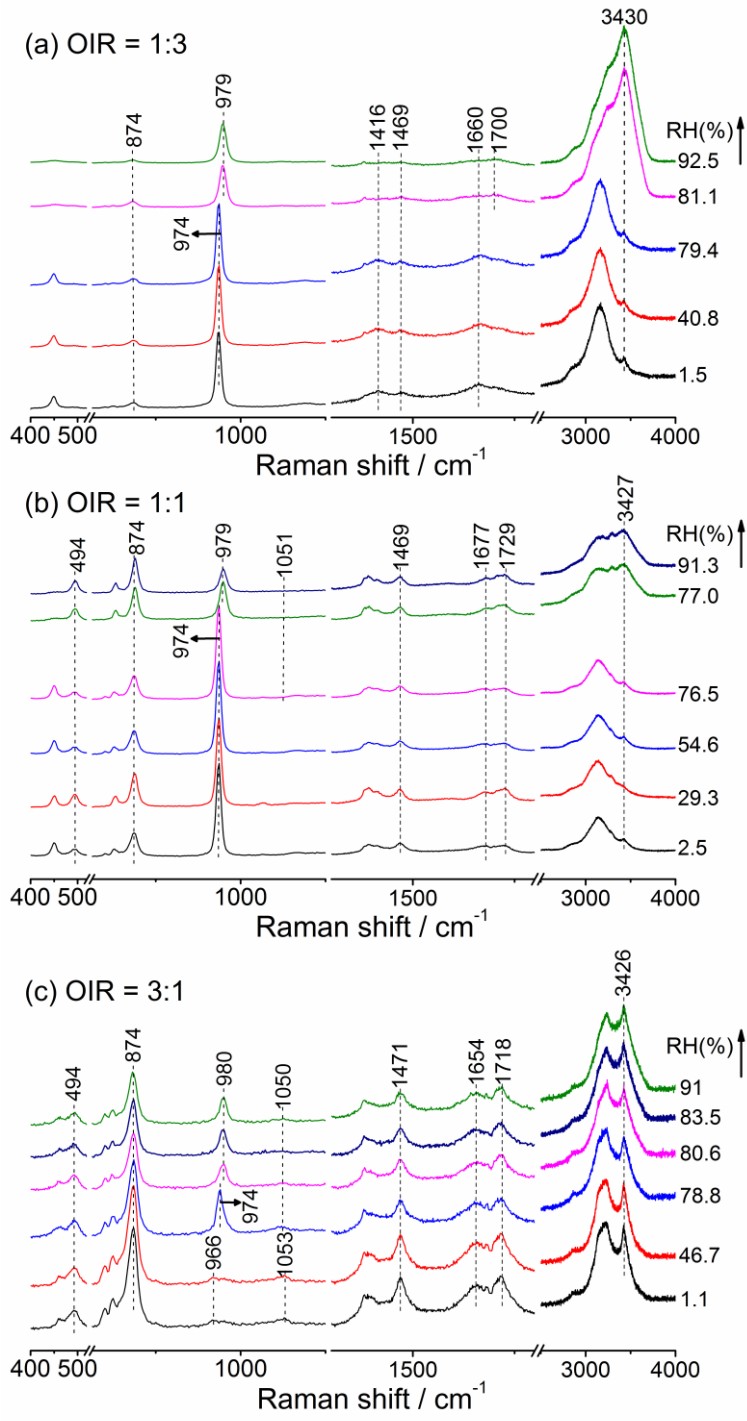

**Figure 4.** Raman spectra of mixed oxalic acid/ammonium sulfate droplets with OIRs of (a) 1:3, (b)

1:1 and (c) 3:1 at various RH values during the hydration process.

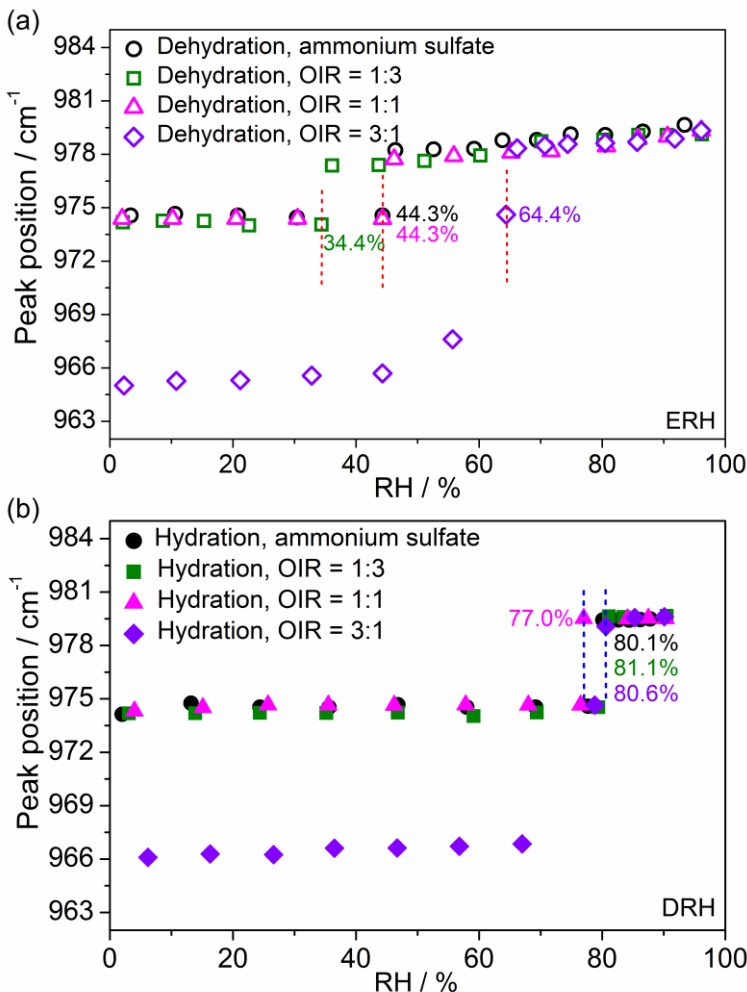

**Figure 5.** The peak position of the $\nu_1$-$SO_4^{2-}$ peak of mixed OA/AS particles and pure AS particles at various RHs during the (a) dehydration and (b) hydration process. The red and blue dashed lines indicate the ERH and DRH, respectively.

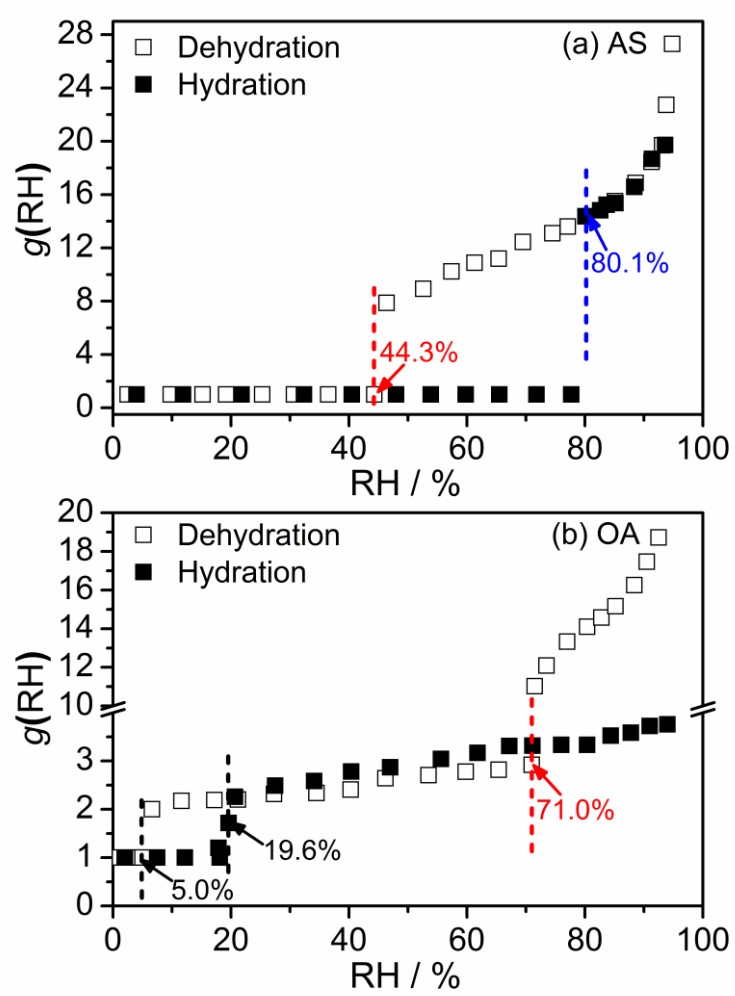

Figure 6. Hygroscopicity of (a) AS and (b) OA as a function of RH. The red and blue dashed lines

indicate the ERH and DRH, respectively. The black lines show phase transition point for the

transformation between oxalic acid dihydrate and anhydrous oxalic acid.

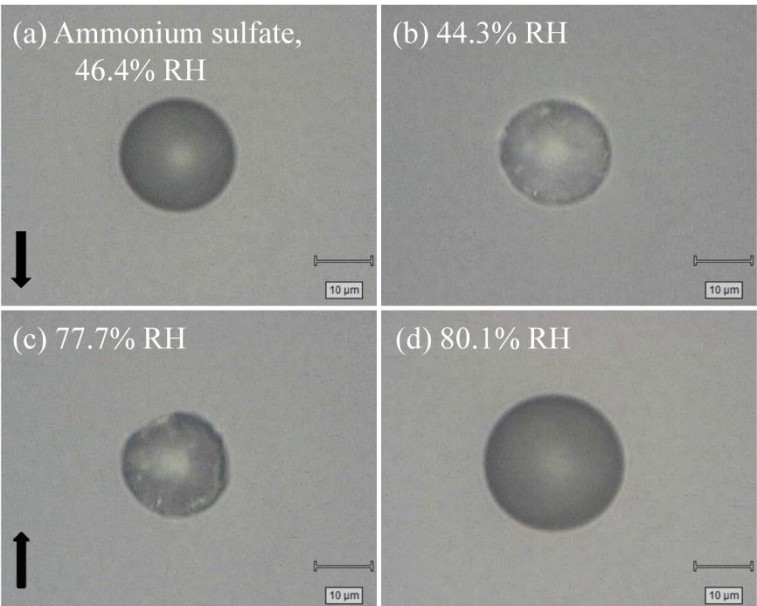

**Figure 7.** Optical micrographs of the ammonium sulfate particle at the phase change points. Dehydration process: (a) 46.4% RH and (b) 44.3% RH. Hydration process: (c) 77.7% RH and (d) 80.1% RH.

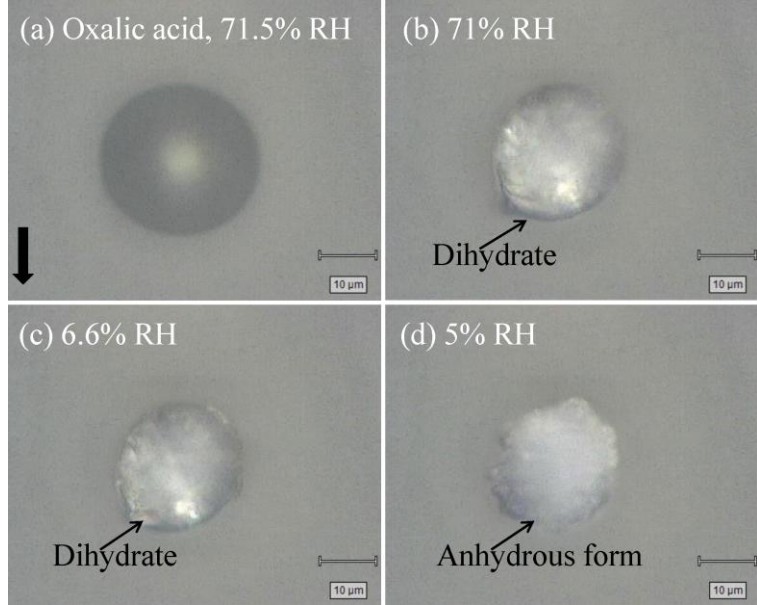

**Figure 8.** Optical micrographs of the oxalic acid particle at (a) 71.5% RH, (b) 71% RH, (c) 6.6% RH and (d) 5% RH during the dehydration process, respectively.

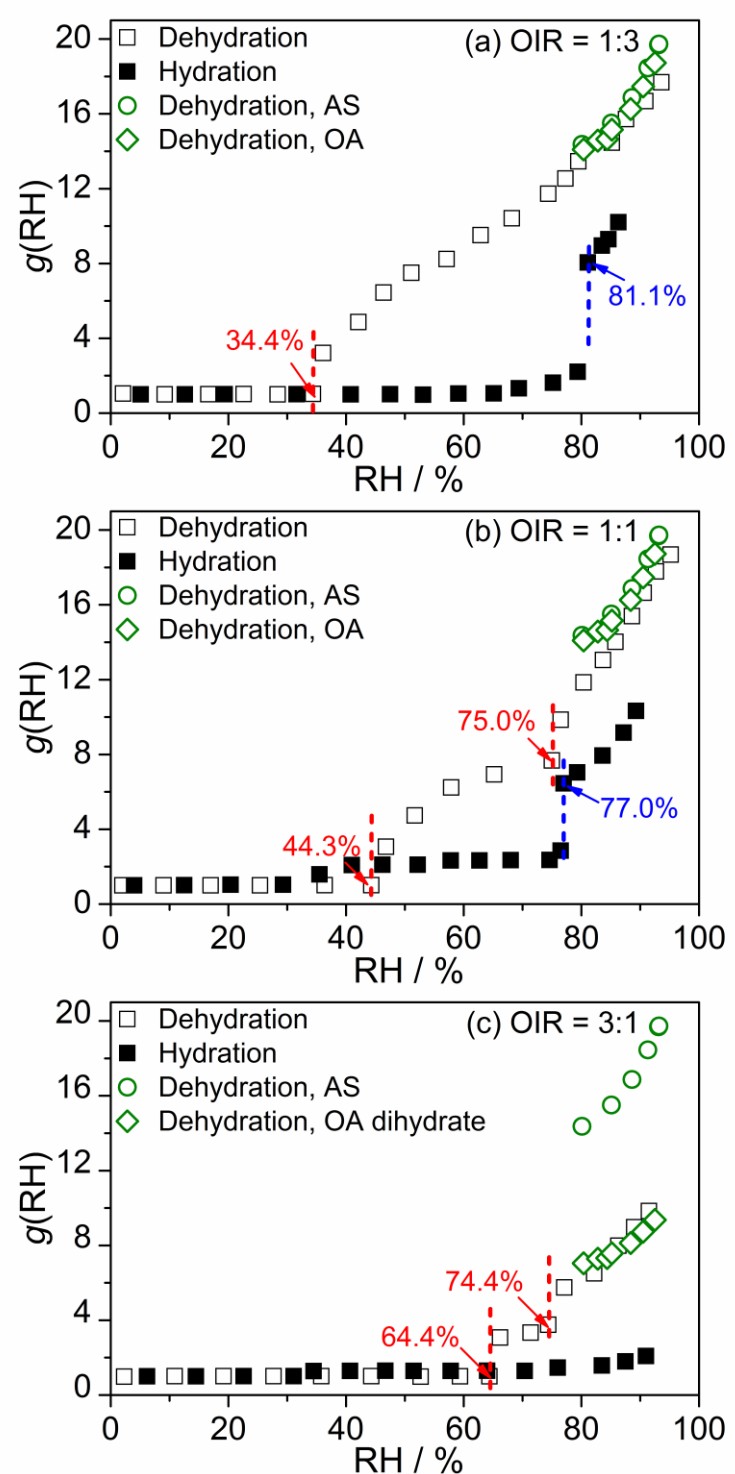

**Figure 9.** Hygroscopicity of OA/AS mixtures with OIRs of (a) 1:3, (b) 1:1 and (c) 3:1 as a function

of RH. The red and blue dashed lines indicate the ERH and DRH, respectively. In panels (a) and (b),

Raman growth factors of pure AS and OA above 80% RH in the dehydration process are also

included for comparisons. In the panel (c), Raman growth factors of pure AS and OA dihydrate

above 80% RH during the dehydration process are also given for comparisons.

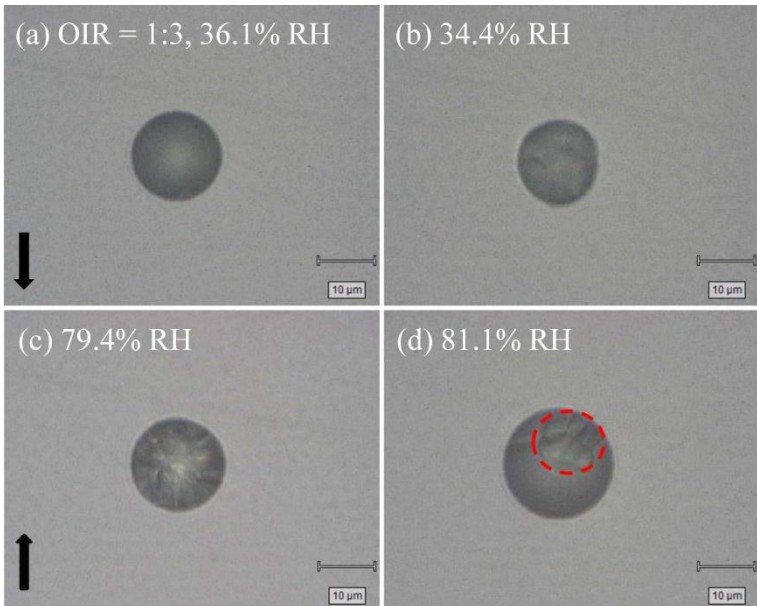

**Figure 10.** Optical micrographs of the mixed oxalic acid/ammonium sulfate particle (OIR = 1:3) at phase change points. Dehydration: (a) 36.1% RH and (b) 34.4% RH. Hydration: (c) 79.4% RH and (d) 81.1% RH. In the image (d), the visual solid in aqueous phase is marked with a red dashed circle.

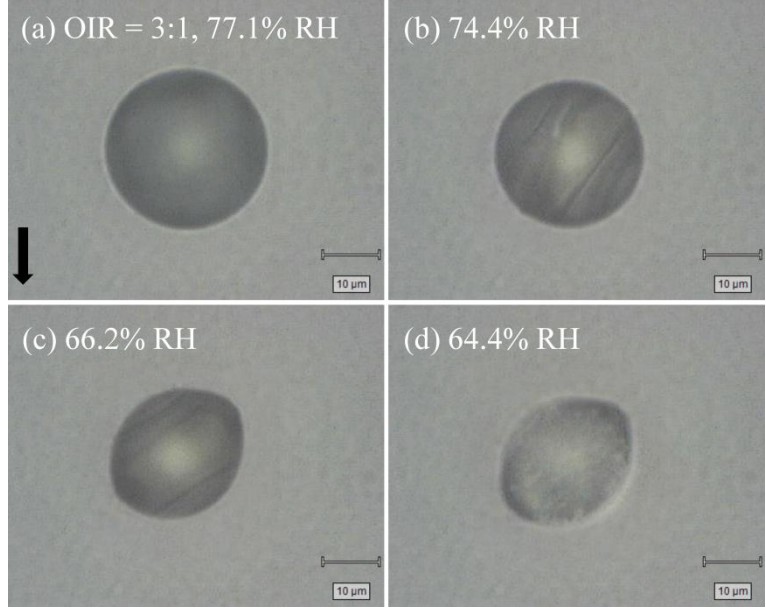

**Figure 11.** Optical micrographs of the mixed oxalic acid/ammonium sulfate particle (OIR = 3:1) at (a) 77.1% RH, (b) 74.4% RH, (c) 66.2% RH and (d) 64.4% RH during the dehydration process, respectively.

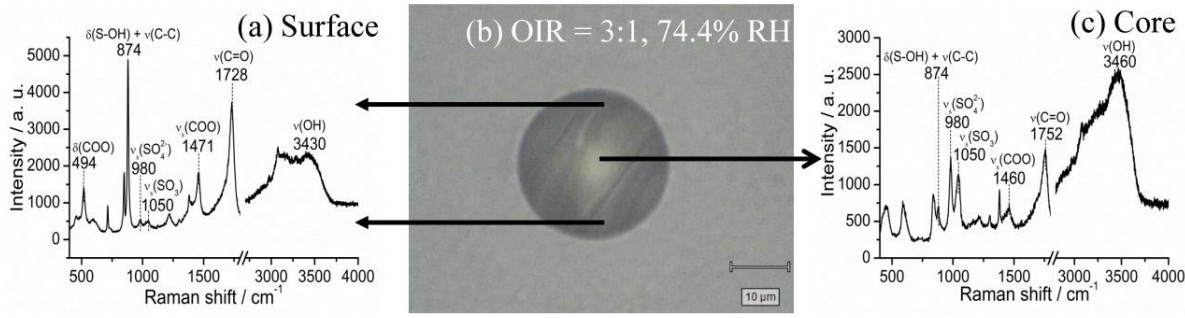

**Figure 12.** The spatial distribution of chemicals within mixed oxalic acid/ammonium sulfate (OIR

= 3:1) particles at 74.4% RH upon dehydration. (a) Raman spectrum acquired on the surface

showing the shell mainly consisting of $NH_4HC_2O_4$. (b) Optical micrograph of a partially effloresced

droplet composed of oxalic acid/ammonium sulfate (OIR = 3:1) mixtures at 74.4% RH upon

dehydration. (c) Raman spectrum obtained at the core of the droplet showing the liquid phase

dominated by oxalic acid and ammonium sulfate.

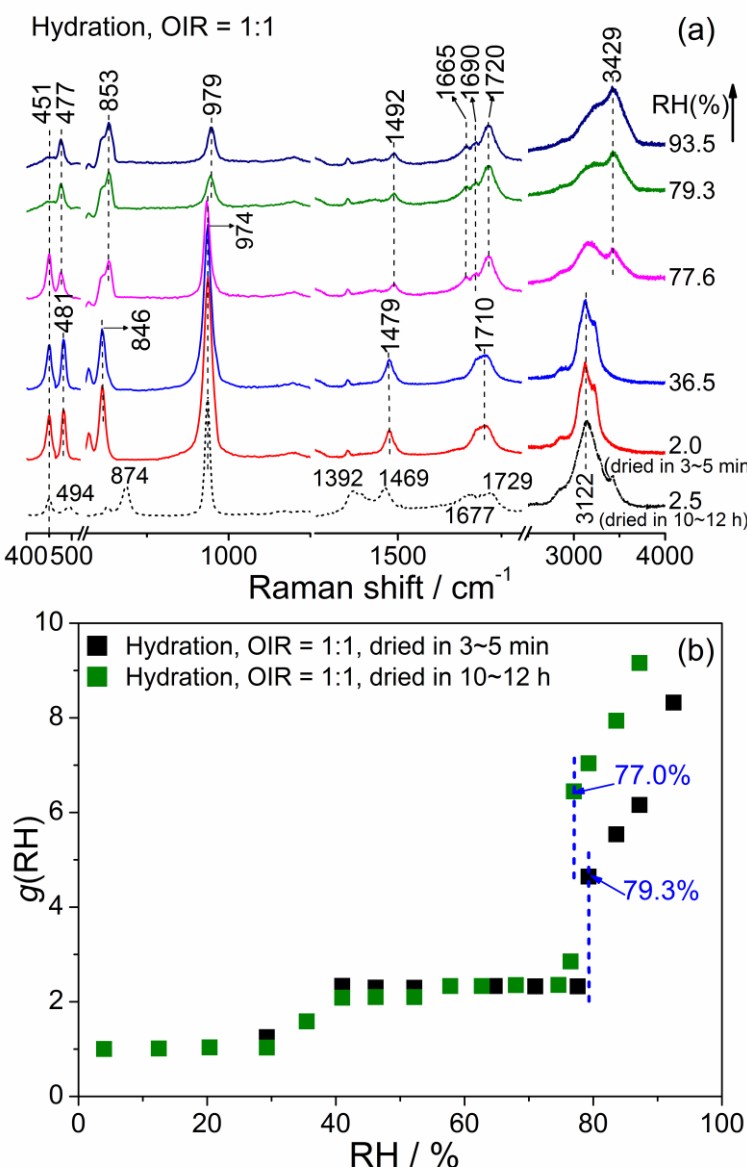

**Figure 13.** (a) Raman spectra of equal molar mixed OA/AS particles after rapid drying process at various RH values upon hydration. The Raman spectrum (black short dash) at 2.5% RH obtained from the slow drying process is also given for comparisons. (b) Deliquescence curve of OA/AS mixtures with an OIR of 1:1. The hygroscopic curve (olive line) of particles after slow drying process is also included for comparisons. The blue dashed lines indicate the DRH.

