# Peer review of "Hygroscopic behavior and chemical composition evolution of"

_Atmospheric Chemistry and Physics, 2017_

## Referee Comment (RC1) · Anonymous Referee #1 · 12 Jun 2017

In this work, the authors measured the hygroscopic phase transitions of organic-sulfate particles. Specifically, pure oxalic acid, pure ammonium sulfate, and mixtures of both were studied using Raman microspectroscopy. Shifts in Raman spectral bands at specific relative humidities marked both deliquescence and efflorescence. The deliquescence relative humidity (DRH) and efflorescence relative humidity (ERH) of oxalic acid (OA) and ammonium (AS) sulfate agree with previous literature with one exception: the ERH of OA was higher (77%) than literature values ($\sim$55%). Mixtures of OA with AS exhibited different hygroscopic behavior than pure substances, which depended on the molar ratio of OA to AS. Increasing the molar ratio of OA to AS increased the mixture's ERH. Furthermore, at the highest OA to AS molar ratio (3:1), aqueous chemical

reactions in the particle during drying formed nonhygroscopic products. These non-hygroscopic products, such as NH4HC2O2 and residual OA, did not deliquesce upon rehumidification.

While this paper uses sound techniques, cites relevant literature, and is well written, it ultimately fails to address relevant scientific questions within the scope of ACP. The reviewer admits this work presents two new pieces of novel data. First, the ERH of the studied mixtures was dependent on the molar ratio of oxalic acid to ammonium sulfate. Second, aqueous reactions in drying droplets of OA and AS can produce non-hygroscopic products. This novel data, however, is not discussed within the framework of our current knowledge in the literature. Thus, it is not clear to the reviewer how atmospherically relevant or important this work is.

This reviewer suggests two pathways to increase the efficacy of this work. In one pathway, the authors would expand this work—especially the discussion section. Currently, the discussion section contains no effort to frame the results of this study into the existing literature. In the second pathway, the authors could to submit to a technical journal that does not emphasize "studies with general implications for atmospheric science." In addition to this very general comment, several general comments and specific comments are outlined below. Technical comments, however, are omitted in this initial review.

General Comments:

Page 5, line 15: In general, the reviewer feels that the authors did not take advantage of the microscope in their experiment. Do the authors know the contact angle of water on their Raman substrate? If so, the physical growth factor of a spherically equivalent drop could be determined; this measurement would greatly increase confidence in the spectroscopic growth factor measurement. A physical growth factor measurement could also help explain the low-RH results in Figure 4b, where it is unclear if OA shrinks when it transitions from its dihydrate form to it anhydrous form.

Page 7, Line 1: Do the authors have an image of the effloresced particle to affirm that the $\nu(SO_4^{2-})$ peak shift corresponds with a hygroscopic phase change?

Page 8, Line 2: Since multiple components are crystallizing, can the authors take advantage of the high spatial resolution of Raman microscopy to tell if there is a spatial distribution of chemicals? These results would explain if components are efflorescing in specific order and, consequently, if effloresced components are heterogeneously nucleating other components.

Specific Comments:

Page 3, Line 16: Is there a reference for the reactions of organic acids with mineral salts, chloride salts, nitrate salts, and ammonium and amines?

Page 4, Line 21: What was the dry diameter of these particles?

Page 5, Line 8: What is the numerical aperture of the 50x objective?

Page 5, Line 12: Why was 40 minutes chosen for the equilibration time? Do the authors have spectral evidence of this equilibration (perhaps from the area under the OH water peak?)

Page 7, Line 3: It is unclear from the text if 874 cm-1 corresponds to only HSO4- or both HSO4- and HC2O4-. The reviewer suggests this be clarified.

Page 9, Line 11: The statement "likely due to drop size, substrate, and experimental methods" is vague. Can the authors be more specific about the cause of OA's high ERH in this study?

Page 9, Line 16: Do the authors believe that 77% is the true ERH of OA, or that heterogeneous nucleation is occurring? If the latter, the reviewer suggests that the authors refrain from using the phrase "ERH of OA" hereafter.

Page 12, Line 8: Do the "rapidly-dried" particles look physically different than the "regularly-dried" particles? Furthermore, do the rapidly-dried particles have a different ERH? This could help discern the underlying mechanism of efflorescence.

---

## Referee Comment (RC2) · Anonymous Referee #2 · 13 Jun 2017

In this paper the authors study the hygroscopic properties and phase transformations of mixed oxalic acid (OA) and ammonium sulfate (AS) particles using confocal Raman spectroscopy. In particular the authors study three different OA/AS ratios: 1:3, 1:1 and 3:1 at room temperature. Pure OA crystalizes into the dihydrate at 77% RH and converts into the dehydrated form at 5% RH. Pure AS particles deliquesce at ~80% RH and effloresce at ~ 44% RH (validating the Raman technique). The observed ERH for the mixed particles ranged from ~34 to 64% RH. The ERH increases as the amount of OA in the mixtures increase. The DRH for the mixed particles ranged from ~77 to 81% RH. The authors claim the formation of different compounds in OA/AS particles during a slow drying process affects the DRH during a subsequent hydration process.

[Figure]

I have many of the same comments as Reviewer #1. In particular revising the discussion section to include atmospheric relevance is crucial before final publication. After revision, I believe the manuscript represents a contribution to scientific progress within the scope of ACP. The scientific approach and methods are valid. I recommend publication in ACP after the authors address the concerns of the reviewers.

Comments:

1) I was wondering if the authors considered referencing and discussing Amundson et al. (2007) which provides a sulfate/ammonium/oxalic acid phase diagram.

Amundson, N.R., Caboussat, A., He, J.W., Martynenko, A.V., and Seinfeld, J.H., A phase equilibrium model for atmospheric aerosols containing inorganic electrolytes and organic compounds (UHAERO), with application to dicarboxylic acids, J. Geohphys. Res., 112, D24S13, 2007.

2) Page 4, line 19: How did the authors create 30-40 micron particles with a syringe? This procedure needs to be explained better. Also, how was the environment of the particles maintained at 95% RH after injection? Aren't the particles subjected to the environment in the room which is surely less than 95% RH? Are 30-40 micron particles relevant in the atmosphere?

3) Page 5, line11: The authors state that the particles were equilibrated with water vapor for 40 minutes at a given RH value and they state that the slow dehydration process occurred in the time scale of hours. Why was the time scale of 40 minutes chosen? Why not 30 minutes or 60 minutes? Is 40 minutes the amount of time for the Raman spectrum to remain constant?

4) Page 5, equation 1: I understand the equation for the growth factor but when the authors create the hygroscopic growth curve is the growth factor an average of many particles or only 1 particle?

5) Figure 1a: This is actually a problem I have with all the figures showing Raman

spectra. There are just too many peak assignments and it clutters the figures up. Can the authors remove any peak assignments that don't illustrate the point of the figure? For example, in the OA dehydration process the peak at 1689 cm-1 is obviously important because that's the peak associated with the dihydrate. That peak should clearly be highlighted. Also can the authors remove any of the Raman spectra that don't highlight something interesting happening? All that's happening is the water peaks are getting smaller. Also, the oxalic acid dihydrate spectrum at the bottom of Figure 1a looks like it is part of the dehydration process. It took me a little bit of time to figure out that the spectrum wasn't part of the dehydration process in the figure. I understand the importance of this spectrum but can it be boxed in or something so the reader doesn't think it is part of the dehydration process?

6) Page 6, line 5: This comment is associated with the comment above. Again, there are too many peak assignments in the text. It clutters the paragraph up. Focus on the most important peaks.

7) General comment: I think it would be interesting to see pictures of the particles during the hydration and the dehydration process. Do the authors have pictures of the particles they could associate with the Raman spectra? The reason I bring this up is Wise et al. (2012) found that when aqueous sodium chloride particles effloresced at low temperatures the dihydrate formed. The morphology of those particles was different than the morphology of the dehydrated form. I am also wondering if the authors could physically see evidence of $NH_4HSO_4$ or $NH_4HC_2O_4$ they claim to see spectral evidence of on page 7, line 4. Additionally can the authors see any coatings they argue are present on page 11, line 11?

Wise, M.E., K.J. Baustian, T. Koop, M.A. Freedman, E. J. Jensen and M.A. Tolbert, Depositional ice nucleation onto hydrated NaCl particles: A new mechanism for ice formation in the troposphere, Atmos. Chem. and Phys., 12, 1121-1134, 2012

8) Page 7, line 10: There needs to be an arrow in the equation not an equal sign.

9) Page 10, line 5: Again, I think pictures of the particles might help strengthen the case for water uptake prior to deliquescence. The authors should be able to see the particles gain water prior to full deliquescence. I am now wondering if the authors could create hygroscopic growth curves utilizing the physical size of the particles and if that correlates with the Raman growth factors.

10) General comment: Can the authors comment on the applicability of the data at temperatures lower than room temperature. Obviously, in the atmosphere the particles are going to experience temperatures much lower than room temperature.

11) Page 5, line 25: Why did the authors decide to put the mixed hydration Raman spectra in the supplemental section? Surely, this data is important to the findings described in the paper.

---

## Referee Comment (RC3) · Anonymous Referee #3 · 26 Jun 2017

The authors presented a laboratory work on the hygroscopic growth and phase transitions of oxalic acid (OA), ammonium sulfate (AS), and their mixed particles. The growth factor and the phase transition of deliquescence and efflorescence were determined using the spectra collected by confocal Raman spectroscopy at room temperature. It is showing that the particles with different mixing ratios showed different hygroscopicity during the hydration and dehydration cycles. At higher OA/AS ratio, the dehydration process produced less hygroscopic organic salt, such as $NH_4HC_2O_2$, from in particle phase reaction within the aqueous droplet as it loses water. In addition, the manuscript shows the possible effects on the growth factor by the different drying rates. The manuscript also provides explanation for the discrepancy on the ERH of

[Figure]

OA compared to the previous studies. This study provides a set of valuable data for the hygroscopicity of model particles generated in the lab. This work demonstrates the effects of aqueous phase reaction on particle hygroscopicity during dehydration which was overlooked in the past. There is a quite important implication to atmospheric chemistry. It is recommended for publication after a minor revision. Please see the following comments which the authors may want to consider in the revision.

Minor comments:

1. P1, L17, L18, "aerosol" refers to the mixture of particle and gases. It is suggested to change the "aerosol" to "particle".

2. P1, L28, how do you define "the partial deliquescence relative humidity"?

3. P3, L23, this statement is not clear, in the previous sentences the authors showed that there are several studies on the OA/AS system. Please provide additional information or references to support this statement.

4. P4, L19, it is not clear how the authors would be able to prepare the 30-40 um aqueous particles with a syringe.

5. P4, L25, If the temperature accuracy is 0.7 K, the uncertainty of RH at 297 K and 95% should be 4%. How the sample temperature is controlled during the experiments?

6. P5, L25, I also suggested to move the Raman spectra to the main text.

7. P6, L11-12, it is not clear to me that how oxalic acid dihydrate can be converted to anhydrous form at these experimental conditions? How long it will take for such process and is it atmospheric relevant?

8. P7, L10, it is the reaction, not an equation.

9. P9, L6-11, the explanation for the discrepancy on the ERH of OA compared to the previous studies should be carefully addressed.

10. P11, L10-12, as suggested by the previous reviewers, it may be more straightforward if the authors can provide optical images to show the phase transitions. For this possible evidence on the coating of less hygroscopic materials, it may be easy to just provide Raman spectral at different location of particles or compositional mapping with the imaging mode.

11. P11, L28-29, it is not clear how the RH is controlled during the 10-12h experimental period, stepwise or continuously? What is the variation of sample temperature during this period?

12. Figure 4 and 5, It is suggested to compare the experimental results with model estimation, such as E-AIM, ZSR, or AIOMFAC. For example, the E-AIM model (http://www.aim.env.uea.ac.uk/aim/aim.php) includes the dissociation equilibrium for some organic/inorganic systems. The oxalic acid is included in current E-AIM. What would E-AIM predict and how does that compare with your experimental data? This can not only serve as validation of the determined Raman growth factor but may also provide additional insides to the effects of reactions on particle's hygroscopicity.

---

## Author Comment (AC1) · 4 Sep 2017

**Author's Response**

**Response to Referee #1:**

We are grateful for the reviewer's comments. Those comments are all valuable and helpful for improving our paper. Our response to the comments and changes to the manuscript are included below. We repeat the specific points raised by the reviewer in bold font, followed by our response in italic font. The pages numbers and lines mentioned below are consistent with those in the Atmospheric Chemistry and Physics Discussions (ACPD) paper.

**While this paper uses sound techniques, cites relevant literature, and is well written, it ultimately fails to address relevant scientific questions within the scope of ACP. The reviewer admits this work presents two new pieces of novel data. First, the ERH of the studied mixtures was dependent on the molar ratio of oxalic acid to ammonium sulfate. Second, aqueous reactions in drying droplets of OA and AS can produce nonhygroscopic products. This novel data, however, is not discussed within the framework of our current knowledge in the literature. Thus, it is not clear to the reviewer how atmospherically relevant or important this work is.**

**This reviewer suggests two pathways to increase the efficacy of this work. In one pathway, the authors would expand this work especially the discussion section. Currently, the discussion section contains no effort to frame the results of this study into the existing literature. In the second pathway, the authors could to submit to a technical journal that does not emphasize "studies with general implications for atmospheric science." In addition to this very general comment, several general comments and specific comments are outlined below. Technical comments, however, are omitted in this initial review.**

*Reply: According to the reviewer's suggestion, we expand relevant discussion on our results within the framework of the existing literature to highlight atmospheric relevance.*

[revised manuscript text omitted]

**Related changes in the revised manuscript:**

**Page 11 line 18-29: The sentences from line 18 to 29 are replaced by** *"The observed efflorescence relative humidity (ERH) for mixed droplets was dependent on the molar ratio of oxalic acid to ammonium sulfate. The mixed OA/AS droplets with an OIR of 1:3 are observed to effloresce completely at 34.4 ± 2.0% RH relative to ERH of pure AS (44.3 ± 2.5%) or OA (77 ± 2.5%). It can be seen that AS as a major fraction of the particle does not promote the heterogeneous nucleation of OA. Meanwhile, the crystallization of AS is also influenced due to the presence of OA. The similar phenomenon was also observed for malonic acid/ammonium sulfate mixtures with minor organic content (Braban and Abbatt, 2004; Parsons et al., 2004). Braban and Abbatt (2004) found that the ERH of malonic acid/ammonium sulfate mixed particles was considerably decreased compared to that of pure ammonium sulfate for mass fractions of malonic acid less than 0.3. They concluded that the presence of ammonium sulfate in the supersaturated droplet could exert the extra barrier to nucleation of malonic acid crystals rather than play the role of a heterogeneous nucleation site. As for 1:3 OA/AS mixed droplets, ammonium sulfate may also inhibit the nucleation of oxalic acid at relatively high RH. With decreasing RH, aqueous oxalic acid could enhance the viscosity of the droplet due to hydrogen bond interactions (Mikhailov et al., 2009), thus limiting the nucleation of ammonium sulfate and resulting in a lower ERH with respect to the value of pure AS (Parsons et al., 2004). In the case of mixed OA/AS droplets with an OIR of 1:1 and 3:1, the $NH_4HC_2O_4$ formed at ~75% RH upon dehydration likely acts as a heterogeneous nucleus for crystallization of other components, which increases full efflorescence point of mixed particles. One study indicated that Aldrich humic acid sodium salt (NaHA) could also promote the ERH of ammonium sulfate (Badger et al., 2006). Similar to oxalic acid, succinic acid and adipic acid have a high deliquescence point and low solubility. However, it has been found that the efflorescence point of ammonium sulfate in mixed particles is not elevated even when the content of succinic acid or adipic acid is not less than 50% by mass or mole fractions (Ling and Chan, 2008; Yeung et al., 2009; Laskina et al., 2015). In contrast to ammonium sulfate particles containing succinic acid or adipic acid, our results*

*suggest that the addition of oxalic acid into ammonium sulfate droplets may trigger partial and full crystallisation of aerosols at relatively higher RH upon dehydration due to $NH_4HC_2O_4$ product acting as an effective nucleus.*

*During the deliquescence process, the OA/AS mixed particles with an OIR of 1:3 and 1:1 exhibit a slightly lower deliquescence point than that of pure ammonium sulfate, consistent with previous observations of effects of crystalline oxalic acid on deliquescence transition of ammonium sulfate (Brooks et al., 2002; Wise et al., 2003; Jing et al., 2016). It should be noted that prior literature result also showed that continuous or smooth water uptake from low RH was observed for particles composed of AS and OA with a mass ratio of 1.5:1 due to the fact that after drying processing oxalic acid existing in an amorphous or liquid-like state prevented nucleation of ammonium sulfate even under dry conditions (Prenni et al., 2003). In the present study, water uptake by the OA/AS mixed particles at high RH upon hydration is dramatically lower than that upon dehydration and significantly decreased with elevated OA content. This phenomenon distinguishes from hygroscopic characteristic of typical water-soluble mixtures in literatures. It has been found that hydration growth curve and dehydration growth curve are typically merged above deliquescence point for mixed systems containing inorganic salts and water-soluble organic compounds (Choi and Chan, 2002; Chan and Chan, 2003; Gysel et al., 2004; Clegg and Seinfeld, 2006; Sjogren et al., 2007; Pope et al., 2010; Ghorai et al., 2014; Estillore et al., 2016). In this study, Raman spectra and micrograph suggest the presence of solid $NH_4HC_2O_4$ and residual solid OA at high RH should be responsible for the decreased water uptake during the hydration process. In contrast, Prenni et al. (2003) reported that the hygroscopic growth of OA/AS mixed particles remained unchanged at 90% RH with OA mass fraction ranging from 0.01 to 0.4. In addition, they also found that water uptake after deliquescence was well described by the model method assuming complete dissolution of OA in aqueous phase as well as no interactions between OA and AS, which was also observed by Jing et al. (2016) using the HTDMA. The previous HTDMA studies for OA/AS mixed particles indicate no composition change and no specific interactions existing between OA and AS (Prenni et al., 2003; Jing et al., 2016). However, it should be noted that the HTDMA studies did not perform measurements for the dehydration process such that aerosols underwent rapid drying on the time scale of seconds, i.e., the total residence time for transformation of droplets into dry particles in the drying section*

*of HTDMA is typically tens of seconds (Prenni et al., 2003; Jing et al., 2016), much shorter than that (10 ~ 12 h) in our study. In the HTDMA experiments, the combination of faster drying and smaller particles with submicron size implies that the aqueous phase obtained higher supersaturations than in our present study (Rosenoern et al., 2008), leading to less dissociation of oxalic acid and thus less $HC_2O_4^-$ formed in the droplets as well as the inhibited formation of $NH_4HC_2O_4$. The fast evaporation of water from the surface of an aqueous droplet upon rapid drying could result in a higher surface concentration of solutes than the slow drying process (Treuel et al., 2011). The higher surface concentration of oxalic acid corresponds to less formation and hence decreased supersaturation of $HC_2O_4^-$. Due to the dependence of nucleation rate on the extent of supersaturation, it can be expected that the nucleation of $NH_4HC_2O_4$ is suppressed within OA/AS mixed droplets undergoing rapid drying.".*

**Page 12 line 20: "4 Conclusions" is changed into "4 Conclusions and atmospheric implications".**

**Page 13 line 9-20: This paragragh is replaced by** *"The prior hygroscopic studies suggest that crystallization of internally mixed ammonium sulfate/dicarboxylic acid particles may lead to the formation of trace organic salt. Lightstone et al. (2000) estimated that approximately 2% of the initial succinic acid may form ammoniated succinate within mixed ammonium nitrate/succinic acid particles during the efflorescence process. Ling and Chan (2008) inferred that crystallization of ammonium sulfate/succinic acid droplets likely generated metastable organic salt based on change in the Raman peak form of succinic acid. Braban and Abbatt (2004) reported that $NH_4HSO_4$ and ammoniated malonate were likely generated upon crystallization of mixed ammonium sulfate/malonic acid particles. However, due to the trace amount of organic salt below Raman or infrared detection limit, they found no apparent influence of organic salt formed upon dehydration on the water uptake or phase change of mixed particles. In contrast, our results indicate that the chemical processing upon drying of droplets containing OA and AS influences efflorescence transition and water uptake of mixed aerosols during the humidity cycle by modifying particulate component.*

*Our results highlight the atmospheric importance of dicarboxylic acid−ammonium sulfate interactions in aerosol aqueous chemistry. Such chemical processing upon drying of aerosols comprised of organic acid/$(NH_4)_2SO_4$ mixtures may enhance the acidity of aqueous phase in the*

*intermediate RH due to the transformation of (NH₄)₂SO₄ into NH₄HSO₄. These experiments also imply that the chemical reaction between aqueous $(NH_4)_2SO_4$ and oxalic acid upon slow dehydration is a possible formation pathway for the low-volatility oxalate in ambient particles, which could enhance partitioning of dicarboxylic acids to aqueous particles with the presence of ammonium sulfate (Yli-Juuti et al., 2013; Hakkinen et al., 2014). It has been reported that the aerosol aqueous processing within organic acid/AS mixtures partly contributes to enhanced loadings of secondary organic aerosol (SOA) from biogenic precursors (Hoyle et al., 2011). Compared to aqueous processing such as condensed phase acid-catalyzed reactions relevant to formation of organosulfates, the contribution of other aerosol processing containing organic salt formation to SOA burden likely becomes important under less acidic condition. Formation of low-solubility organic salts from aqueous processing within aerosols alters particle-phase component and thus modifies aerosol's hygroscopicity, optical properties and chemical reactivity. Our findings provide fundamental insight into effects of drying conditions (drying rate or time) on formation of organic salt from reactions of organic acids with inorganic salts in particle phase under ambient RH conditions. Overall, a better understanding of the chemical interactions between species in a multicomponent system during the humidity cycle is critical for the accurate modeling efforts of aerosol phase behavior in thermodynamic models.".*

**General Comments:**

**Page 5, line 15: In general, the reviewer feels that the authors did not take advantage of the microscope in their experiment. Do the authors know the contact angle of water on their Raman substrate? If so, the physical growth factor of a spherically equivalent drop could be determined; this measurement would greatly increase confidence in the spectroscopic growth factor measurement. A physical growth factor measurement could also help explain the low-RH results in Figure 4b, where it is unclear if OA shrinks when it transitions from its dihydrate form to it anhydrous form.**

*Reply: We thank the reviewer for the suggestion. We have no contact angle data for droplets on our substrate. In fact, the spectra methods have been proved to be sensitive and reliable for study of aerosol hygroscopicity including phase transition and water uptake (Cziczo et al., 1997; Cziczo and Abbatt, 2000; Braban et al., 2003; Brooks et al., 2003; Braban and Abbatt, 2004;*

*Garland et al., 2005; Badger et al., 2006; Liu et al., 2008a; Liu et al., 2008b; Yeung et al., 2009; Minambres et al., 2010; Yeung and Chan, 2010; Ghorai et al., 2014; Laskina et al., 2015; Zawadowicz et al., 2015).*

*As shown in Figure R1, the size of an effloresced oxalic acid particle remains almost unchanged when oxalic acid dihydrate is transformed into anhydrous form. However, the corresponding Raman spectra indicate the changes in crystal water of OA particles. The other studies using infrared spectrometer and vapor sorption analyzer also observed the transition between anhydrous oxalic acid and dihydrate based on water mass changes in solid OA particles (Braban et al., 2003; Ma et al., 2013). Since size-based hygroscopicity is sensitive to particle geometry, the size growth factor of particles without a compact structure may not reflect the actual changes in water mass due to morphology effects (Piens et al., 2016). It seems that the structure of anhydrous OA particle is not as compact as that of dihydrate, seen in Figure R1. Thus, the loss of crystal water results in no obvious change in particle size. Overall, the spectra method is advantageous for probing the hygroscopic behavior of atmospheric particles with irregular morphologies.*

***Related changes included in the revised manuscript:***

*Figure R1 is supplemented in the main text.* ***Page 9, Line 5:*** *"As shown in Fig. 4b, the measured ERH of OA is $77 \pm 2.5\%$ RH"* ***is revised to*** *"As shown in Fig. 6b and 8, the measured ERH of OA is $77 \pm 2.5\%$ RH".* ***Page 9, Line 26: We add*** *"It seems that the structure of anhydrous OA particle is not as compact as that of dihydrate, seen in Fig. 8. Thus, the loss of crystal water results in no obvious change in particle size.".*

[Figure]

***Figure R1.*** *Optical micrographs of the oxalic acid particle at (a) 77.3% RH, (b) 77% RH, (c) 6.6% RH and (d) 5% RH during the dehydration process, respectively.*

**Page 7, Line 1: Do the authors have an image of the effloresced particle to affirm that the $v(SO_4^{2-})$ peak shift corresponds with a hygroscopic phase change?**

***Reply:*** *As seen in Figure R2, the crystallization of OA/AS particles (OIR = 1:3) occurs at 34.4% RH, corresponding with the Raman peak shift of $v_s(SO_4^{2-})$ from 979 $cm^{-1}$ to 974 $cm^{-1}$ at the same RH. The previous studies have also applied the abrupt shift in characteristic peak position to indicate phase transition of ammonium sulfate during the hygroscopic process (Braban and Abbatt, 2004; Ling and Chan, 2008; Yeung et al., 2009; Yeung and Chan, 2010).*

***Related changes in the revised manuscript:***

*Figure R2 is supplemented in the main text.* ***Page 7, Line 1:*** *The sentence "At 34.4% RH, the shift of $v_s(SO_4^{2-})$ peak from 979 $cm^{-1}$ to 974 $cm^{-1}$ indicates the crystallization of AS."* ***is revised to*** *"At 34.4% RH, the shift of $v_s(SO_4^{2-})$ peak from 979 $cm^{-1}$ to 974 $cm^{-1}$ indicates the crystallization of AS, as also seen in Fig. 10b.".* ***Page 8, Line 6: We add*** *"The previous studies have also applied the abrupt shift in characteristic peak position to indicate phase transition of ammonium sulfate during the hygroscopic process (Braban and Abbatt, 2004; Ling and Chan, 2008; Yeung et al., 2009).".*

[Figure]

***Figure R2.*** *Optical micrographs of mixed oxalic acid/ammonium sulfate particles (OIR = 1:3) at phase change points. Dehydration: (a) 36.1% RH and (b) 34.4% RH. Hydration: (c) 79.4% RH and (d) 81.1% RH. In the image (d), the visual solid in aqueous phase is marked with a red dashed circle.*

**Page 8, Line 2: Since multiple components are crystallizing, can the authors take advantage of the high spatial resolution of Raman microscopy to tell if there is a spatial distribution of chemicals? These results would explain if components are efflorescing in specific order and, consequently, if effloresced components are heterogeneously nucleating other components.**

***Reply:*** *We appreciate the reviewer's suggestion. Figure R3 presents the spatial distribution of chemicals within mixed OA/AS (OIR = 3:1) particles at 74.4% RH. The characteristic peak of 980 $cm^{-1}$, 1050 $cm^{-1}$ and 1471 $cm^{-1}$ is assigned to $SO_4^{2-}$, $HSO_4^-$ and $HC_2O_4^-$, respectively. The sharp absorption at 874 $cm^{-1}$ and obvious peak at 1471 $cm^{-1}$ indicate the abundant content of $NH_4HC_2O_4$. The comparison of characteristic peaks between inner and outer phase reveals that the major component on the surface of a mixed OA/AS (OIR = 3:1) particle is $NH_4HC_2O_4$. In contrast to the surface, the obvious features of 980 $cm^{-1}$ and 1050 $cm^{-1}$ at the core of the particle suggest that $(NH_4)_2SO_4$ and $NH_4HSO_4$ mainly exist in the inner aqueous phase. During the dehydration process, crystalline $NH_4HC_2O_4$ in the outer phase acts as heterogeneous nucleus, leading to the crystallization of oxalic acid dihydrate, $(NH_4)_2SO_4$ and $NH_4HSO_4$ in the inner phase.*

***Related changes in the revised manuscript:***

*Figure R3 is added into the text.* **Page 11, Line 4:** *The sentence "The crystallization of NH$_4$HC$_2$O$_4$ may act as crystallization nuclei for NH$_4^+$, HSO$_4^-$ and OA in the mixed droplets to form NH$_4$HSO$_4$ crystal and oxalic acid dihydrate."* **is changed into** *"Figure 12 presents the spatial distribution of chemicals within mixed OA/AS (OIR = 3:1) particles at 74.4% RH. The characteristic peak of 980 cm$^{-1}$, 1050 cm$^{-1}$ and 1471 cm$^{-1}$ is assigned to SO$_4^{2-}$, HSO$_4^-$ and HC$_2$O$_4^-$, respectively. The sharp absorption at 874 cm$^{-1}$ and obvious peak at 1471 cm$^{-1}$ indicate the abundant content of NH$_4$HC$_2$O$_4$. The comparison of characteristic peaks between inner and outer phase reveals that the major component on the surface of a mixed OA/AS (OIR = 3:1) particle is NH$_4$HC$_2$O$_4$. In contrast to the surface, the obvious features of 980 cm$^{-1}$ and 1050 cm$^{-1}$ at the core of the particle suggest that (NH$_4$)$_2$SO$_4$ and NH$_4$HSO$_4$ mainly exist in the inner aqueous phase. During the dehydration process, crystalline NH$_4$HC$_2$O$_4$ in the outer phase acts as the heterogeneous nucleus, leading to the crystallization of oxalic acid dihydrate, (NH$_4$)$_2$SO$_4$ and NH$_4$HSO$_4$ in the inner phase.".*

[Figure]

**Figure R3.** *The spatial distribution of chemicals within the mixed oxalic acid/ammonium sulfate (OIR = 3:1) particle at 74.4% RH upon dehydration. (a) Raman spectrum acquired on the surface showing the shell mainly consisting of NH$_4$HC$_2$O$_4$. (b) Optical micrograph of a partially effloresced droplet composed of oxalic acid/ammonium sulfate (OIR = 3:1) mixtures at 74.4% RH upon dehydration. (c) Raman spectrum obtained at the core of the droplet showing the liquid phase dominated by oxalic acid and ammonium sulfate.*

**Specific Comments:**

**Page 3, Line 16: Is there a reference for the reactions of organic acids with mineral salts, chloride salts, nitrate salts, and ammonium and amines?**

**Reply:** *We add several references for the reactions of organic acids with mineral salts, chloride*

*salts, nitrate salts, and ammonium and amines.*

***Related changes in the revised manuscript:***

***Page 3, Line 13:*** *The sentence "Field measurements have observed the formation of low-volatility organic salts in atmospheric particles due to the reactions of organic acids with mineral salts, chloride salts, nitrate salts, ammonium and amines."* ***is revised to*** *"Field measurements have observed the formation of low-volatility organic salts in atmospheric particles due to the reactions of organic acids with mineral salts, chloride salts, nitrate salts, ammonium and amines (Sullivan and Prather, 2007; Laskin et al., 2012; Wang and Laskin, 2014; Smith et al., 2010)."*

**Page 4, Line 21: What was the dry diameter of these particles?**

***Reply:*** *The dry diameter of these particles after efflorescence ranged from 10 to 20 μm.*

***Related changes in the revised manuscript:***

***Page 4, Line 21: We add the sentence*** *"The dry size of these particles after efflorescence ranged from 10 to 20 μm.".*

**Page 5, Line 8: What is the numerical aperture of the 50x objective?**

***Reply:*** *The numerical aperture of the 50× objective is 0.75.*

***Related changes in the revised manuscript:***

***Page 5, Line 8:*** *The sentence "Then, spectroscopic measurements were made on droplets observed by using the Leica DMLM microscope with a 50× objective lens."* ***is revised to*** *"Then, spectroscopic measurements were made on droplets observed by using the Leica DMLM microscope with a 50× objective (0.75 numerical aperture)."*

**Page 5, Line 12: Why was 40 minutes chosen for the equilibration time? Do the authors have spectral evidence of this equilibration (perhaps from the area under the OH water peak?)**

***Reply:*** *We used intensity ratios of the water peak (3430 cm$^{-1}$) to the sulfate peak (980 cm$^{-1}$) to test the equilibration time of droplets at the given RH. Our results indicate that the intensity ratios remain almost unchanged after 20 min for a 30 μm droplet. To achieve the full equilibration for*

*particles with size range studied, the droplets were equilibrated with water vapor at an ambient relative humidity for about 40 min. After 40 min, the Raman spectra in our experiment remain constant. Yeung et al. (2009) determined the equilibration time of at least 15 min for a 20-30 μm ammonium sulfate droplet based on the intensity ratio of the water peak (3430 cm$^{-1}$) to the sulfate peak (980 cm$^{-1}$) obtained by micro-Raman spectroscopy. They also found the equilibration time was longer for the same-sized particles containing organics.*

***Related changes in the revised manuscript:***

***Page 5, Line 11:*** *The sentence "The particles were equilibrated with water vapor at a given RH for about 40 min." **is revised to** "The particles were equilibrated with water vapor at a given RH for about 40 min, during which the intensity ratios of the water peak (3430 cm$^{-1}$) to the sulfate peak (980 cm$^{-1}$) remained constant.".*

**Page 7, Line 3: It is unclear from the text if 874 cm$^{-1}$ corresponds to only HSO$_4^-$ or both HSO$_4^-$ and HC$_2$O$_4^-$. The reviewer suggests this be clarified.**

***Reply:*** *The band centred at 874 cm$^{-1}$ is contributed by both HSO$_4^-$ and HC$_2$O$_4^-$. Dawson et al. (1986) reported the absorption of vibrational mode (δ(S-OH)) of HSO$_4^-$ ion from NH$_4$HSO$_4$ occurred at 869 cm$^{-1}$. The absorption of ν(C-C) of HC$_2$O$_4^-$ in crystal was observed at 879 cm$^{-1}$ by Shippey (1979). Thus, the peak centred at 874 cm$^{-1}$ corresponds to both HSO$_4^-$ and HC$_2$O$_4^-$.*

***Related changes in the revised manuscript:***

***Page 7, Line 3:*** *The sentence "A new band centered at 874 cm$^{-1}$ corresponds to the vibrational mode (δ(S-OH)) of HSO$_4^-$ ion from NH$_4$HSO$_4$ and the HC$_2$O$_4^-$ ion vibrating (Irish and Chen, 1970; Dawson et al., 1986; Villepin and Novak, 1971; Shippey, 1979)," **is revised to** "A new band centered at 874 cm$^{-1}$ corresponds to combination bands of the vibrational mode (δ(S-OH)) of HSO$_4^-$ ion from NH$_4$HSO$_4$ (Dawson et al., 1986) and HC$_2$O$_4^-$ ion vibrating (Shippey, 1979)".*

**Page 9, Line 11: The statement "likely due to drop size, substrate, and experimental methods" is vague. Can the authors be more specific about the cause of OA's high ERH in this study?**

***Reply:*** *We thank the reviewer for the helpful suggestion. After revisiting our explanation carefully, we give a more specific one as follows. The discrepancy on the ERH of OA compared to that*

*reported by Peng et al. (2001) is likely due to the effects of substrate and sample purity. The size of dry particles ranging from 10 to 20 μm in our experiment is consistent with observation using EDB by Peng et al. (2001), which eliminates the influence of particle size. The substrate supporting droplets may promote the heterogeneous nucleation of oxalic acid while the levitated droplets in EDB study can avoid induced nucleation by the substrate. Ghorai et al. (2014) also reported the potential effects of substrate on the efflorescence transition of NaCl/dicarboxylic acid mixed particles. In addition, The OA purity in our study is 99.0% lower than that of 99.5% in study by Peng et al. (2001). Thus, trace amounts of impurities in OA droplets acting as a heterogeneous nucleus could contribute to crystallization and result in a higher ERH of OA. Due to the effects of substrate and sample purity, the heterogeneous nucleation should be responsible for the discrepancy on the observed ERH of OA.*

***Corresponding changes in the revised manuscript:***

***Page 9, Line 10-17:*** *the sentence "The discrepancies between this study and that by Peng et al. (2001) is likely due to the effects of droplet size, substrate and experimental method. According to classical nucleation theory, the probability of the formation of the critical nucleus is proportional to the particle volume (Martin, 2000; Parsons et al., 2006). Considering that the droplet size in our study was approximately 1-2 times larger than that observed by Peng et al. (2001), the droplets deposited on the substrate in our experiment may promote the heterogeneous nucleation while the levitated droplets using EDB can dispel the heterogeneous nucleation. Thus, the ERH of OA obtained in our study is higher than the observation of Peng et al. (2001)." **is revised to** "The discrepancy on the ERH of OA compared to that reported by Peng et al. (2001) is likely due to the effects of substrate and sample purity. The size of dry particles ranging from 10 to 20 μm in our experiment is consistent with observation using EDB by Peng et al. (2001), which eliminates the influence of particle size. The substrate supporting droplets may promote the heterogeneous nucleation of oxalic acid while the levitated droplets in EDB study can avoid induced nucleation by the substrate. Ghorai et al. (2014) also reported the potential effects of substrate on the efflorescence transition of NaCl/dicarboxylic acid mixed particles. In addition, The OA purity in our study is 99.0% lower than that of 99.5% in study by Peng et al. (2001). Thus, trace amounts of impurities in OA droplets acting as a heterogeneous nucleus could contribute to crystallization and result in a higher ERH of OA. Due to the effects of substrate and sample purity, the*

*heterogeneous nucleation should be responsible for the discrepancy on the observed ERH of OA.".*

**Page 9, Line 16: Do the authors believe that 77% is the true ERH of OA, or that heterogeneous nucleation is occurring? If the latter, the reviewer suggests that the authors refrain from using the phrase "ERH of OA" hereafter.**

*Reply: Yes, we determine that 77% is the true ERH of pure OA from the Raman spectra. As stated in the initial manuscript, the Raman spectra indicated OA droplet was converted into oxalic acid dihydrate at 77% RH during the dehydration process. In addition, the images of OA particles upon dehydration also show the full efflorescence of OA occurs at 77% RH, seen in Figure R1.*

***Related changes in the revised manuscript:***

*Figure R1 is supplemented in the main text. **Page 9, Line 5:** "As shown in Fig. 4b, the measured ERH of OA is 77 ± 2.5% RH" **is revised to** "As shown in Fig. 6b and 8, the measured ERH of OA is 77 ± 2.5% RH".*

**Page 12, Line 8: Do the "rapidly-dried" particles look physically different than the "regularly-dried" particles? Furthermore, do the rapidly-dried particles have a different ERH? This could help discern the underlying mechanism of efflorescence.**

*Reply: As shown in Figure R4, the morphology of rapidly-dried OA/AS particles with equal molar ratio could not be obviously distinguished from that of regularly-dried particles. However, the spectra evidence has shown significant compositional difference between the two kinds of particles. We observed one-step efflorescence of rapidly-dried particles (1:1, molar ratio) occurred at 47% ± 2.5% RH, compared to the two-step efflorescence of slowly-dried particles occurring at 75% and 44.3% RH, respectively.*

***Related changes in the revised manuscript:***

***Page 12, line 2: We add** "We observed one-step efflorescence of rapidly-dried particles (1:1, molar ratio) occurred at 47% ± 2.5% RH, compared to the two-step efflorescence of slowly-dried particles occurring at 75% and 44.3% RH, respectively.".*

[Figure]

*Figure R4. Optical micrographs of equal molar mixed oxalic acid/ammonium sulfate particles after (a) rapid drying at 2% RH and (b) slow drying at 2.5% RH, respectively.*

---

## Author Comment (AC2) · 4 Sep 2017

**Author's Response**

**Response to Referee #2:**

We are grateful for the reviewer's comments. Those comments are all valuable and helpful for improving our paper. Our response to the comments and changes to the manuscript are included below. We repeat the specific points raised by the reviewer in bold font, followed by our response in italic font. The pages numbers and lines mentioned below are consistent with those in the Atmospheric Chemistry and Physics Discussions (ACPD) paper.

**I have many of the same comments as Reviewer #1. In particular revising the discussion section to include atmospheric relevance is crucial before final publication. After revision, I believe the manuscript represents a contribution to scientific progress within the scope of ACP. The scientific approach and methods are valid. I recommend publication in ACP after the authors address the concerns of the reviewers.**

*Reply: We thank the reviewer for the comments. We have supplemented atmospheric relevance as follows.*

*Related changes in the revised manuscript:*

*Page 12 line 20: "4 Conclusions" is changed into "4 Conclusions and atmospheric implications".*

***Page 13 line 9-20: This paragragh is replaced by** "The prior hygroscopic studies suggest that crystallization of internally mixed ammonium sulfate/dicarboxylic acid particles may lead to the formation of trace organic salt. Lightstone et al. (2000) estimated that approximately 2% of the initial succinic acid may form ammoniated succinate within mixed ammonium nitrate/succinic acid particles during the efflorescence process. Ling and Chan (2008) inferred that crystallization of ammonium sulfate/succinic acid droplets likely generated metastable organic salt based on change in the Raman peak form of succinic acid. Braban and Abbatt (2004) reported that $NH_4HSO_4$ and ammoniated malonate were likely generated upon crystallization of mixed ammonium sulfate/malonic acid particles. However, due to the trace amount of organic salt below Raman or infrared detection limit, they found no apparent influence of organic salt formed upon dehydration on the water uptake or phase change of mixed particles. In contrast, our results indicate that the chemical processing upon drying of droplets containing OA and AS influences efflorescence transition and water uptake of mixed aerosols during the humidity cycle*

*by modifying particulate component.*

*Our results highlight the atmospheric importance of dicarboxylic acid−ammonium sulfate interactions in aerosol aqueous chemistry. Such chemical processing upon drying of aerosols comprised of organic acid/(NH₄)₂SO₄ mixtures may enhance the acidity of aqueous phase in the intermediate RH due to the transformation of $(NH_4)_2SO_4$ into $NH_4HSO_4$. These experiments also imply that the chemical reaction between aqueous $(NH_4)_2SO_4$ and oxalic acid upon slow dehydration is a possible formation pathway for the low-volatility oxalate in ambient particles, which could enhance partitioning of dicarboxylic acids to aqueous particles with the presence of ammonium sulfate (Yli-Juuti et al., 2013; Hakkinen et al., 2014). It has been reported that the aerosol aqueous processing within organic acid/AS mixtures partly contributes to enhanced loadings of secondary organic aerosol (SOA) from biogenic precursors (Hoyle et al., 2011). Compared to aqueous processing such as condensed phase acid-catalyzed reactions relevant to formation of organosulfates, the contribution of other aerosol processing containing organic salt formation to SOA burden likely becomes important under less acidic condition. Formation of low-solubility organic salts from aqueous processing within aerosols alters particle-phase component and thus modifies aerosol's hygroscopicity, optical properties and chemical reactivity. Our findings provide fundamental insight into effects of drying conditions (drying rate or time) on formation of organic salt from reactions of organic acids with inorganic salts in particle phase under ambient RH conditions. Overall, a better understanding of the chemical interactions between species in a multicomponent system during the humidity cycle is critical for the accurate modeling efforts of aerosol phase behavior in thermodynamic models.".*

**1) I was wondering if the authors considered referencing and discussing Amundson et al. (2007) which provides a sulfate/ammonium/oxalic acid phase diagram.**

**Reply:** *Thanks for the reviewer's suggestion. Amundson et al. (2007) presented a phase partitioning model (UHAERO) for mixtures of inorganic electrolytes and dicarboxylic acids. They assumed that solid oxalic acid was the only organic solid that could occur in the sulfate/ammonium/oxalic acid system. The limitations on the simple assumption for this system were a result of the lack of available thermodynamic data. Amundson et al. (2007) considered the incorporation of organic salts was crucial in the modeling of hygroscopic properties as well as*

*multistage growth of organic/inorganic mixtures.*

***Related changes in the revised manuscript:***

***P4, L2: We add*** *"Due to the lack of available thermodynamic data, the aerosol thermodynamic models typically assume that upon dehydration dicarboxylic acid could only form organic solid without the organic salt in the inorganic electrolyte/dicarboxylic acid system (Clegg and Seinfeld, 2006; Amundson et al., 2007). Thus, the incorporation of organic salts formed from interactions between inorganic salts and organic acids is crucial in the modeling of hygroscopic properties of mixed organic/inorganic particles.".*

**2) Page 4, line 19: How did the authors create 30-40 micron particles with a syringe? This procedure needs to be explained better. Also, how was the environment of the particles maintained at 95% RH after injection? Aren't the particles subjected to the environment in the room which is surely less than 95% RH? Are 30-40 micron particles relevant in the atmosphere?**

***Reply:*** *The sample solution was discharged from a syringe. Then, residual solution in the syringe was pushed rapidly to generate aerosol droplets spraying onto a PTFE substrate fixed to the bottom of the sample cell. Then, the sample cell was promptly sealed by a transparent polyethylene film. The RH in the sample cell was regulated by nitrogen streams consisting of a mixture of water-saturated $N_2$ and dry $N_2$ at controlled flow rates. At ~ 95% RH, the droplets with a diameter of 30~40 microns detected by an optical microscope (50× objective, 0.75 numerical aperture) were selected to acquire the Raman spectra. The droplet size of 30-40 micron in our study falls into the size range of cloud droplets (less than 50 μm).*

***Related changes in the revised manuscript:***

***P4, L19:*** *the sentence "Using a syringe, droplets from the solutions were injected onto the polytetrafluorethylene (PTFE) film fixed to the bottom of the sample cell. The diameters of these droplets ranged from 30 to 40 μm at ~ 95% RH. Then, the sample cell was sealed by a transparent polyethylene film and the RH in the sample cell was regulated by nitrogen streams consisting of a mixture of water-saturated $N_2$ and dry $N_2$ at controlled flow rates."* ***is revised to*** *"The sample solution was discharged from a syringe. Then, residual solution in the syringe was pushed rapidly to generate aerosol droplets spraying onto a polytetrafluorethylene (PTFE)*

*substrate fixed to the bottom of the sample cell. Then, the sample cell was promptly sealed by a transparent polyethylene film. The RH in the sample cell was regulated by nitrogen streams consisting of a mixture of water-saturated $N_2$ and dry $N_2$ at controlled flow rates. At ~ 95% RH, the droplets with a diameter of 30~40 microns detected by an optical microscope (50× objective, 0.75 numerical aperture) were selected to acquire the Raman spectra. The dry size of these particles after efflorescence ranged from 10 to 20 μm.".*

**3) Page 5, line 11: The authors state that the particles were equilibrated with water vapor for 40 minutes at a given RH value and they state that the slow dehydration process occurred in the time scale of hours. Why was the time scale of 40 minutes chosen? Why not 30 minutes or 60 minutes? Is 40 minutes the amount of time for the Raman spectrum to remain constant?**

***Reply:*** *We used intensity ratios of the water peak (3430 cm$^{-1}$) to the sulfate peak (980 cm$^{-1}$) to test the equilibration time of droplets at the given RH. Our results indicate that the intensity ratios remain almost unchanged after 20 min for a 30 μm droplet. To achieve the full equilibration for particles with size range studied, the droplets were equilibrated with water vapor at an ambient relative humidity for about 40 min. After 40 min, the Raman spectra in our experiment remain constant. Yeung et al. (2009) determined the equilibration time of at least 15 min for a 20-30 μm ammonium sulfate droplet based on the intensity ratio of the water peak (3430 cm$^{-1}$) to the sulfate peak (980 cm$^{-1}$) obtained by micro-Raman spectroscopy. They also found the equilibration time was longer for the same-sized particles containing organics.*

***Related changes in the revised manuscript:***

***Page 5, Line 11:*** *The sentence "The particles were equilibrated with water vapor at a given RH for about 40 min." **is revised to** "The particles were equilibrated with water vapor at a given RH for about 40 min, during which the intensity ratios of the water peak (3430 cm$^{-1}$) to the sulfate peak (980 cm$^{-1}$) remained constant.".*

**4) Page 5, equation 1: I understand the equation for the growth factor but when the authors create the hygroscopic growth curve is the growth factor an average of many particles or only 1 particle?**

*Reply:* *Multiple particles (three or four) were selected to acquire the Raman spectra through each humidity cycle. Thus, the hygroscopic growth curve is derived from average growth factors of multiple particles. Each measurement for one particle was also repeated at least three times.*

***Related changes in the revised manuscript:***

***Page 5, Line 13: We add the sentence*** *"Multiple particles (three or four) were selected to acquire the Raman spectra through each humidity cycle."*

***Page 5, Line 20:*** *The sentence "Hygroscopic growth curves are acquired by plotting the Raman growth factor as a function of RH."* ***is changed into*** *"Hygroscopic growth curves are acquired by plotting the average Raman growth factor of duplicate particles as a function of RH.".*

**5) Figure 1a: This is actually a problem I have with all the figures showing Raman spectra. There are just too many peak assignments and it clutters the figures up. Can the authors remove any peak assignments that don't illustrate the point of the figure? For example, in the OA dehydration process the peak at 1689 cm$^{-1}$ is obviously important because that's the peak associated with the dihydrate. That peak should clearly be highlighted. Also can the authors remove any of the Raman spectra that don't highlight something interesting happening? All that's happening is the water peaks are getting smaller. Also, the oxalic acid dihydrate spectrum at the bottom of Figure 1a looks like it is part of the dehydration process. It took me a little bit of time to figure out that the spectrum wasn't part of the dehydration process in the figure. I understand the importance of this spectrum but can it be boxed in or something so the reader doesn't think it is part of the dehydration process?**

*Reply:* *Thanks for the reviewer's suggestion. We remove some minor peak assignments in the figures. To avoid misunderstanding, the oxalic acid dihydrate spectrum at the bottom of Figure 2a is indicated by a black dash line in the modified version.*

***Related changes in the revised manuscript:***

*Figure R1 is the Figure 2 in the modified version. The other figures are also modified according to the reviewer's suggestion.*

[Figure]

**Figure R1.** *Raman spectra of oxalic acid droplets during the (a) dehydration process and (b) hydration process. In panel (a), the black dashed line indicates the spectrum of pure $H_2C_2O_4 \cdot 2H_2O$ particles with the peak height of $v(OH)$ located at 3433 $cm^{-1}$ scaled by a factor of 1/6.*

**6) Page 6, line 5: This comment is associated with the comment above. Again, there are too many peak assignments in the text. It clutters the paragraph up. Focus on the most important peaks.**

*Reply: According to reviewer's suggestion, we remove some minor peak assignments in the text.*

***Related changes in the revised manuscript:***

***Page 6, line 5:*** *"As seen in Fig. 1a, the feature bands for OA droplets are observed at 457, 845, 1460, 1636, 1750 and 3433 $cm^{-1}$ at 92.5% RH. At lower RH around 77% (Fig. 1a, magenta line), these bands shift to 477, 855, 1490, 1627, 1737, 3433 and 3474 $cm^{-1}$, and a new band at 1689*

*cm$^{-1}$ occurs, which is entirely consistent with the spectrum of oxalic acid dihydrate (Fig. 1a, black line)." is changed into "As seen in Fig. 2a, the feature bands for OA droplets are observed at 1460, 1750 and 3433 cm$^{-1}$ at 92.5% RH. At lower RH around 77% (Fig. 2a, magenta line), these bands shift to 1490, 1737, 3433 and 3474 cm$^{-1}$, and a new band at 1689 cm$^{-1}$ occurs, which is entirely consistent with the spectrum of oxalic acid dihydrate (Fig. 2a, black dashed line).". Similar modifications in other places are also made.*

**7) General comment: I think it would be interesting to see pictures of the particles during the hydration and the dehydration process. Do the authors have pictures of the particles they could associate with the Raman spectra? The reason I bring this up is Wise et al. (2012) found that when aqueous sodium chloride particles effloresced at low temperatures the dihydrate formed. The morphology of those particles was different than the morphology of the dehydrated form. I am also wondering if the authors could physically see evidence of NH$_4$HSO$_4$ or NH$_4$HC$_2$O$_4$ they claim to see spectral evidence of on page 7, line 4. Additionally can the authors see any coatings they argue are present on page 11, line 11?**

*Reply: We thank for the reviewer's suggestion. Although we have pictures of the particles associated with the Raman spectra, we could not distinguish the morphology of dry particles between dihydrate form and anhydrous one except for oxalic acid particles (Figure R2). Also, on page 7, line 4, NH$_4$HSO$_4$ or NH$_4$HC$_2$O$_4$ could not be identified from the solid phase by visual inspection, seen in Figure R3 (b). On page 11, line 11, NH$_4$HC$_2$O$_4$ coatings on 3:1 OA/AS particles were formed during the dehydration process, seen in Figure R4.*

***Related changes in the revised manuscript:***

*Some important pictures of the particles during the humidity cycle are supplemented into the modified version.*

*The picture of 3:1 OA/AS particles with NH$_4$HC$_2$O$_4$ coatings formed during the dehydration process is added in the main text.*

[Figure]

***Figure R2.*** *Optical micrographs of the oxalic acid particle at (a) 77.3% RH, (b) 77% RH, (c) 6.6% RH and (d) 5% RH during the dehydration process, respectively.*

[Figure]

***Figure R3.*** *Optical micrographs of the mixed oxalic acid/ammonium sulfate particle (OIR=1:3) upon dehydration: (a) 36.1% RH and (b) 34.4% RH.*

[Figure]

***Figure R4.*** *The spatial distribution of chemicals within mixed oxalic acid/ammonium sulfate (OIR = 3:1) particles at 74.4% RH upon dehydration. (a) Raman spectrum acquired on the surface showing the shell mainly consisting of $NH_4HC_2O_4$. (b) Optical micrograph of a partially effloresced droplet composed of*

*oxalic acid/ammonium sulfate (OIR = 3:1) mixtures at 74.4% RH upon dehydration. (c) Raman spectrum obtained at the core of the droplet showing the liquid phase dominated by oxalic acid and ammonium sulfate.*

**8) Page 7, line 10: There needs to be an arrow in the equation not an equal sign.**

***Reply:*** *P7, L10, we replace an equal sign with an arrow for the reaction.*

**9) Page 10, line 5: Again, I think pictures of the particles might help strengthen the case for water uptake prior to deliquescence. The authors should be able to see the particles gain water prior to full deliquescence. I am now wondering if the authors could create hygroscopic growth curves utilizing the physical size of the particles and if that correlates with the Raman growth factors.**

***Reply:*** *Page 10, line 5, the picture of mixed OA/AS particles with an OIR of 1:3 prior to full deliquescence is given in Figure R5 (c). It can be seen that the size of 1:3 mixed OA/AS particle at 79.4% RH prior to deliquescence appears to be larger than that after complete efflorescence (Figure R5 (b)), suggesting slight water uptake, as also confirmed by the Raman spectrum. Due to the limitation of instrument, the picture resolution is not high enough to help identify distinct liquid water. In addition, the slight water content may exist in the veins and cavities of the particle.*

*Since size-based hygroscopicity is sensitive to particle geometry, physical size of the particles may not reflect the additions of water mass due to morphology effects (Piens et al., 2016). Due to the lack of contact angle data, we cannot create hygroscopic growth curves based on the physical size of the particles. In fact, the spectra method is advantageous for probing the hygroscopic behavior and water content of atmospheric particles with regular or irregular morphologies.*

***Related changes in the revised manuscript:***

*Figure R5 and corresponding descriptions have been supplemented in the text.*

[Figure]

***Figure R5.*** *Optical micrographs of the mixed oxalic acid/ammonium sulfate particle (OIR = 1:3) at phase change points. Dehydration: (a) 36.1% RH and (b) 34.4% RH. Hydration: (c) 79.4% RH and (d) 81.1% RH. In the image (d), the visual solid in aqueous phase is marked with a red dashed circle.*

**10) General comment: Can the authors comment on the applicability of the data at temperatures lower than room temperature. Obviously, in the atmosphere the particles are going to experience temperatures much lower than room temperature.**

***Reply:*** *Zobrist et al. (2006) investigated the heterogeneous freezing points of the aqueous oxalic acid/ammonium sulfate solutions. They found that oxalic acid precipitated as $NH_4H_3(C_2O_4)_2$ $2H_2O$ in the mixed solution to act as a heterogeneous ice nucleus. The crystallization of oxalic acid/ammonium sulfate mixed systems at low temperature may show distinct behaviors relative to room temperature. Thus, we cannot give effective suggestions on applicability of our data at low temperatures.*

**11) Page 5, line 25: Why did the authors decide to put the mixed hydration Raman spectra in the supplemental section? Surely, this data is important to the findings described in the paper.**

***Reply:*** *According to the reviewer's suggestion, we move the mixed hydration Raman spectra in the supplemental section to the text.*

***Related changes in the revised manuscript:***

***P6, L24:*** *The sentence: "The Raman spectra of mixed OA/AS droplets with OIRs of 1:3, 1:1 and 3:1 at various RHs during the dehydration process are depicted in Fig. 2. The corresponding spectra for hydration process are given in Fig. S2 in the Supplement."* ***is revised to*** *"The Raman spectra of mixed OA/AS droplets with OIRs of 1:3, 1:1 and 3:1 at various RHs during the dehydration and hydration process are depicted in Fig. 3 and 4, respectively.".*

**References**

Amundson, N. R., Caboussat, A., He, J. W., Martynenko, A. V., and Seinfeld, J. H.: A phase equilibrium model for atmospheric aerosols containing inorganic electrolytes and organic compounds (UHAERO), with application to dicarboxylic acids, J. Geophys. Res.: Atmos., 112, D24S13, 2007.

Braban, C. F., and Abbatt, J. P. D.: A study of the phase transition behavior of internally mixed ammonium sulfate-malonic acid aerosols, Atmos. Chem. Phys., 4, 1451-1459, 2004.

Clegg, S. L., and Seinfeld, J. H.: Thermodynamic models of aqueous solutions containing inorganic electrolytes and dicarboxylic acids at 298.15 K. 1. The acids as nondissociating components, J. Phys. Chem. A, 110, 5692-5717, 10.1021/jp056149k, 2006.

Hakkinen, S. A. K., McNeill, V. F., and Riipinen, I.: Effect of Inorganic Salts on the Volatility of Organic Acids, Environ. Sci. Technol., 48, 13718-13726, 10.1021/es5033103, 2014.

Hoyle, C. R., Boy, M., Donahue, N. M., Fry, J. L., Glasius, M., Guenther, A., Hallar, A. G., Hartz, K. H., Petters, M. D., Petaja, T., Rosenoern, T., and Sullivan, A. P.: A review of the anthropogenic influence on biogenic secondary organic aerosol, Atmos. Chem. Phys., 11, 321-343, 10.5194/acp-11-321-2011, 2011.

Lightstone, J. M., Onasch, T. B., Imre, D., and Oatis, S.: Deliquescence, efflorescence, and water activity in ammonium nitrate and mixed ammonium nitrate/succinic acid microparticles, J. Phys. Chem. A, 104, 9337-9346, 10.1021/jp002137h, 2000.

Ling, T. Y., and Chan, C. K.: Partial crystallization and deliquescence of particles containing ammonium sulfate and dicarboxylic acids, Journal of Geophysical Research: Atmospheres, 113, 1-15, doi: 10.1029/2008JD009779, 2008.

Piens, D. S., Kelly, S. T., Harder, T. H., Petters, M. D., O'Brien, R. E., Wang, B., Teske, K., Dowell, P., Laskin, A., and Gilles, M. K.: Measuring mass-based hygroscopicity of atmospheric particles through in situ imaging, Environ. Sci. Technol., 50, 5172-5180, doi: 10.1021/acs.est.6b00793, 2016.

Yeung, M. C., Lee, A. K. Y., and Chan, C. K.: Phase transition and hygroscopic properties of internally

mixed ammonium sulfate and adipic acid (AS-AA) particles by optical microscopic imaging and Raman spectroscopy, Aerosol Sci. Technol., 43, 387–399, doi: 10.1080/02786820802672904, 2009.

Yli-Juuti, T., Zardini, A. A., Eriksson, A. C., Hansen, A. M. K., Pagels, J. H., Swietlicki, E., Svenningsson, B., Glasius, M., Worsnop, D. R., Riipinen, I., and Bilde, M.: Volatility of Organic Aerosol: Evaporation of Ammonium Sulfate/Succinic Acid Aqueous Solution Droplets, Environ. Sci. Technol., 47, 12123-12130, 10.1021/es401233c, 2013.

Zobrist, B., Marcolli, C., Koop, T., Luo, B. P., Murphy, D. M., Lohmann, U., Zardini, A. A., Krieger, U. K., Corti, T., Cziczo, D. J., Fueglistaler, S., Hudson, P. K., Thomson, D. S., and Peter, T.: Oxalic acid as a heterogeneous ice nucleus in the upper troposphere and its indirect aerosol effect, Atmos. Chem. Phys., 6, 3115-3129, 2006.

---

## Author Comment (AC3) · 4 Sep 2017

**Author's Response**

**Response to Referee #3:**

We are grateful for the reviewer's comments. Those comments are all valuable and helpful for improving our paper. Our response to the comments and changes to the manuscript are included below. We repeat the specific points raised by the reviewer in bold font, followed by our response in italic font. The pages numbers and lines mentioned below are consistent with those in the Atmospheric Chemistry and Physics Discussions (ACPD) paper.

**The authors presented a laboratory work on the hygroscopic growth and phase transitions of oxalic acid (OA), ammonium sulfate (AS), and their mixed particles. The growth factor and the phase transition of deliquescence and efflorescence were determined using the spectra collected by confocal Raman spectroscopy at room temperature. It is showing that the particles with different mixing ratios showed different hygroscopicity during the hydration and dehydration cycles. At higher OA/AS ratio, the dehydration process produced less hygroscopic organic salt, such as NH4HC2O2, from in particle phase reaction within the aqueous droplet as it loses water. In addition, the manuscript shows the possible effects on the growth factor by the different drying rates. The manuscript also provides explanation for the discrepancy on the ERH of OA compared to the previous studies. This study provides a set of valuable data for the hygroscopicity of model particles generated in the lab. This work demonstrates the effects of aqueous phase reaction on particle hygroscopicity during dehydration which was overlooked in the past. There is a quite important implication to atmospheric chemistry. It is recommended for publication after a minor revision. Please see the following comments which the authors may want to consider in the revision.**

Minor comments:

**1. P1, L17, L18, "aerosol" refers to the mixture of particle and gases. It is suggested to change the "aerosol" to "particle".**

*Reply: Thanks for your suggestion. P1, L17, L18, "aerosol"* **is changed into** *"particle" in the revised version.*

**2. P1, L28, how do you define "the partial deliquescence relative humidity"?**

*Reply: The partial deliquescence relative humidity is used to indicate the RH at which AS deliquesces while other coexisting components in the mixed particles remain solid during the hydration process.*

***Related changes in the revised manuscript:***

***P1, L28,*** *to avoid the misunderstanding, "the partial deliquescence relative humidity (DRH) for mixed OA/AS particles"* ***is revised to*** *"the deliquescence relative humidity (DRH) of AS in mixed OA/AS particles".*

**3. P3, L23, this statement is not clear, in the previous sentences the authors showed that there are several studies on the OA/AS system. Please provide additional information or references to support this statement.**

***Reply:*** *P3, L23, this statement means that the effects of OA on* ***deliquescence*** *behaviors of AS have been extensively investigated (Brooks et al., 2002; Prenni et al., 2003; Wise et al., 2003; Miñambres et al., 2013; Jing et al., 2016), while there is still lack of study on influence of OA on* ***efflorescence*** *behaviors of AS. In the original version, we have used the term "***deliquescence process***" or "***efflorescence process***" to distinguish the studies on hygroscopicity of the OA/AS mixed system.*

**4. P4, L19, it is not clear how the authors would be able to prepare the 30-40 μm aqueous particles with a syringe.**

***Reply:*** *The sample solution was discharged from a syringe. Then, residual solution in the syringe was pushed rapidly to generate aerosol droplets spraying onto a PTFE substrate fixed to the bottom of the sample cell. At ~ 95% RH, the droplets with a diameter of 30~40 microns detected by an optical microscope (50× objective, 0.75 numerical aperture) were selected to acquire the Raman spectra.*

***Related changes in the revised manuscript:***

***P4, L19:*** *The sentence "Using a syringe, droplets from the solutions were injected onto the polytetrafluorethylene (PTFE) film fixed to the bottom of the sample cell. The diameters of these droplets ranged from 30 to 40 μm at ~ 95% RH. Then, the sample cell was sealed by a transparent polyethylene film and the RH in the sample cell was regulated by nitrogen streams*

*consisting of a mixture of water-saturated N$_2$ and dry N$_2$ at controlled flow rates." **is revised to** "The sample solution was discharged from a syringe. Then, residual solution in the syringe was pushed rapidly to generate aerosol droplets spraying onto a polytetrafluorethylene (PTFE) substrate fixed to the bottom of the sample cell. Then, the sample cell was promptly sealed by a transparent polyethylene film. The RH in the sample cell was regulated by nitrogen streams consisting of a mixture of water-saturated N$_2$ and dry N$_2$ at controlled flow rates. At ~ 95% RH, the droplets with a diameter of 30~40 microns detected by an optical microscope (50× objective, 0.75 numerical aperture) were selected to acquire the Raman spectra. The dry size of these particles after efflorescence ranged from 10 to 20 μm.".*

**5. P4, L25, If the temperature accuracy is 0.7 K, the uncertainty of RH at 297 K and 95% should be 4%. How the sample temperature is controlled during the experiments?**

*Reply: We thank for the reviewer's comment. Below 90% RH, the uncertainty of RH at 297 K was less than ±2.5%. We agree that the temperature accuracy of 0.7 K could result in uncertainty of 4% at RH of 95%. The temperature of the sample was maintained at 297 ± 0.5 K by using an automatic thermostat. We would like to add some changes to make it clear.*

*Related changes in the revised manuscript:*

*P4, L24: The sentence "The RH and temperature of the outflow from the sample cell was measured by a humidity/temperature meter (Centertek Center 313) with an accuracy of ±2.5% and ±0.7 K placed near the exit of the sample cell." **is changed into** "The RH and temperature of the outflow from the sample cell was measured by a humidity/temperature meter (Centertek Center 313) with an accuracy of ±2.5% below 90% RH and ±0.7 K placed near the exit of the sample cell.".*

*P4, L26: We add "The temperature accuracy of 0.7 K could result in uncertainty of 4% at RH of 95%. The temperature of the sample was maintained at 297 ± 0.5 K by using an automatic thermostat.".*

**6. P5, L25, I also suggested to move the Raman spectra to the main text.**

*Reply: Thanks for your suggestion. The Raman spectra of AS droplets are moved to the main text.*

*Related changes in the revised manuscript:*

***P5, L24,*** *the sentence "The Raman spectra of AS droplets during the dehydration and hydration process as a function of RH can be found in Fig. S1 (a) and (b) in the supplement, respectively."* ***is revised to*** *"The Raman spectra of AS droplets during the dehydration and hydration process can be found in Fig. 1a and 1b, respectively.".*

**7. P6, L11-12, it is not clear to me that how oxalic acid dihydrate can be converted to anhydrous form at these experimental conditions? How long it will take for such process and is it atmospheric relevant?**

***Reply:*** *In our experiments, oxalic acid particles after efflorescence exist in the form of dihydrate until 6.6% RH, at which the Raman spectrum of dihydrate remains unchanged for 40 min. Once RH decreases to 5%, oxalic acid dihydrate is* ***promptly*** *converted to anhydrous oxalic acid, as seen in Fig. 2a. This conversion only takes a few seconds during our observations. Our results indicate extremely dry conditions may favor the conversion of oxalic acid dihydrate into anhydrous form in the atmospheric environment.*

***Related changes in the revised manuscript:***

*To make it clear,* ***P6, L9, we add*** *"Oxalic acid particles after efflorescence exist in the form of dihydrate until 6.6% RH, at which the Raman spectrum of dihydrate remains unchanged for 40 min.".* ***P6, L9,*** *the sentence "As RH further decreases to ~5.0%, the peaks shift to 482, 828, 845, 1477, 1710, 2587, 2760 and 2909 cm$^{-1}$,"* ***is changed into*** *"Once RH decreases to ~5.0%, the peaks promptly shift to 1477, 1710, 2587, 2760 and 2909 cm$^{-1}$,".*

**8. P7, L10, it is the reaction, not an equation.**

***Reply:*** *P7, L10, we replace the equal sign with an arrow for the reaction.*

**9. P9, L6-11, the explanation for the discrepancy on the ERH of OA compared to the previous studies should be carefully addressed.**

***Reply:*** *We thank the reviewer for the helpful suggestion. After revisiting our explanation carefully, we give a more specific one as follows. The discrepancy on the ERH of OA compared to that reported by Peng et al. (2001) is likely due to the effects of substrate and sample purity. The size of dry particles ranging from 10 to 20 μm in our experiment is consistent with observation using*

*EDB by Peng et al. (2001), which eliminates the influence of particle size. The substrate supporting droplets may promote the heterogeneous nucleation of oxalic acid while the levitated droplets in EDB study can avoid induced nucleation by the substrate. Ghorai et al. (2014) also reported the potential effects of substrate on the efflorescence transition of NaCl/dicarboxylic acid mixed particles. In addition, The OA purity in our study is 99.0% lower than that of 99.5% in study by Peng et al. (2001). Thus, trace amounts of impurities in OA droplets acting as a heterogeneous nucleus could contribute to crystallization and result in a higher ERH of OA. Due to the effects of substrate and sample purity, the heterogeneous nucleation should be responsible for the discrepancy on the observed ERH of OA.*

***Corresponding changes in the revised manuscript:***

***Page 9, Line 10-17:*** *The sentence "The discrepancies between this study and that by Peng et al. (2001) is likely due to the effects of droplet size, substrate and experimental method. According to classical nucleation theory, the probability of the formation of the critical nucleus is proportional to the particle volume (Martin, 2000; Parsons et al., 2006). Considering that the droplet size in our study was approximately 1-2 times larger than that observed by Peng et al. (2001), the droplets deposited on the substrate in our experiment may promote the heterogeneous nucleation while the levitated droplets using EDB can dispel the heterogeneous nucleation. Thus, the ERH of OA obtained in our study is higher than the observation of Peng et al. (2001)." **is revised to** "The discrepancy on the ERH of OA compared to that reported by Peng et al. (2001) is likely due to the effects of substrate and sample purity. The size of dry particles ranging from 10 to 20 μm in our experiment is consistent with observation using EDB by Peng et al. (2001), which eliminates the influence of particle size. The substrate supporting droplets may promote the heterogeneous nucleation of oxalic acid while the levitated droplets in EDB study can avoid induced nucleation by the substrate. Ghorai et al. (2014) also reported the potential effects of substrate on the efflorescence transition of NaCl/dicarboxylic acid mixed particles. In addition, The OA purity in our study is 99.0% lower than that of 99.5% in study by Peng et al. (2001). Thus, trace amounts of impurities in OA droplets acting as a heterogeneous nucleus could contribute to crystallization and result in a higher ERH of OA. Due to the effects of substrate and sample purity, the heterogeneous nucleation should be responsible for the discrepancy on the observed ERH of OA.".*

**10. P11, L10-12, as suggested by the previous reviewers, it may be more straightforward if the authors can provide optical images to show the phase transitions. For this possible evidence on the coating of less hygroscopic materials, it may be easy to just provide Raman spectral at different location of particles or compositional mapping with the imaging mode.**

*Reply: We appreciate the reviewer's comments. The optical images showing the phase transitions have been added in the text. Figure R1 presents the spatial distribution of chemicals within mixed OA/AS (OIR = 3:1) particles at 74.4% RH. The characteristic peak of 980 $cm^{-1}$, 1050 $cm^{-1}$ and 1471 $cm^{-1}$ is assigned to $SO_4^{2-}$, $HSO_4^-$ and $HC_2O_4^-$, respectively. The sharp absorption at 874 $cm^{-1}$ and obvious peak at 1471 $cm^{-1}$ indicate the abundant content of $NH_4HC_2O_4$. The comparison of characteristic peaks between inner and outer phase reveals that the major component on the surface of a mixed OA/AS (OIR = 3:1) particle is $NH_4HC_2O_4$. In contrast to the surface, the obvious features of 980 $cm^{-1}$ and 1050 $cm^{-1}$ at the core of the particle suggest that $(NH_4)_2SO_4$ and $NH_4HSO_4$ mainly exist in the inner aqueous phase. During the dehydration process, crystalline $NH_4HC_2O_4$ in the outer phase acts as a heterogeneous nucleus, leading to the crystallization of oxalic acid dihydrate, $(NH_4)_2SO_4$ and $NH_4HSO_4$ in the inner phase.*

*Related changes included in the revised manuscript:*

*Figure R1 is added into the text. **Page 11, Line 4:** The sentence "The crystallization of $NH_4HC_2O_4$ may act as crystallization nuclei for $NH_4^+$, $HSO_4^-$ and OA in the mixed droplets to form $NH_4HSO_4$ crystal and oxalic acid dihydrate." is changed into "Figure 12 presents the spatial distribution of chemicals within mixed OA/AS (OIR = 3:1) particles at 74.4% RH. The characteristic peak of 980 $cm^{-1}$, 1050 $cm^{-1}$ and 1471 $cm^{-1}$ is assigned to $SO_4^{2-}$, $HSO_4^-$ and $HC_2O_4^-$, respectively. The sharp absorption at 874 $cm^{-1}$ and obvious peak at 1471 $cm^{-1}$ indicate the abundant content of $NH_4HC_2O_4$. The comparison of characteristic peaks between inner and outer phase reveals that the major component on the surface of a mixed OA/AS (OIR = 3:1) particle is $NH_4HC_2O_4$. In contrast to the surface, the obvious features of 980 $cm^{-1}$ and 1050 $cm^{-1}$ at the core of the particle suggest that $(NH_4)_2SO_4$ and $NH_4HSO_4$ mainly exist in the inner aqueous phase. During the dehydration process, crystalline $NH_4HC_2O_4$ in the outer phase acts as the heterogeneous nucleus, leading to the crystallization of oxalic acid dihydrate, $(NH_4)_2SO_4$ and $NH_4HSO_4$ in the inner phase.".*

[Figure]

***Figure R1.*** *The spatial distribution of chemicals within mixed oxalic acid/ammonium sulfate (OIR = 3:1)*
*particles at 74.4% RH upon dehydration. (a) Raman spectrum acquired on the surface showing the shell*
*mainly consisting of NH$_4$HC$_2$O$_4$. (b) Optical micrograph of a partially effloresced droplet composed of*
*oxalic acid/ammonium sulfate (OIR = 3:1) mixtures at 74.4% RH upon dehydration. (c) Raman spectrum*
*obtained at the core of the droplet showing the liquid phase dominated by oxalic acid and ammonium*
*sulfate.*

**11. P11, L28-29, it is not clear how the RH is controlled during the 10-12h experimental
period, stepwise or continuously? What is the variation of sample temperature during this
period?**

***Reply:*** *The RH was decreased stepwise from ~95% to ~0% over 10-12 h during the dehydration*
*process. The decrease rate was typically 5-6 RH/40 min, and the rate remained 2-3 RH/40 min*
*near the phase transition. The temperature of the sample was maintained at 297 ± 0.5 K by using*
*an automatic thermostat. In the Experimental section, we have stated that RH was decreased*
*stepwise during the slow drying process.*

***Related changes included in the revised manuscript:***

***P5, L10,*** *the sentence "Subsequently, the RH was decreased stepwise for dehydration process,*
*and increased from RH < 3% to high RH for hydration process."* ***is revised into*** *"Subsequently,*
*the RH was decreased stepwise for a slow dehydration process, and then increased stepwise from*
*RH < 3% to high RH for a hydration process. The decrease rate was typically 5-6 RH/40 min,*
*and the rate remained 2-3 RH/40 min near the phase transition. The RH was decreased*
*continuously in a few minutes for a rapid dehydration process.".*

**12. Figure 4 and 5, It is suggested to compare the experimental results with model**

**estimation, such as E-AIM, ZSR, or AIOMFAC. For example, the E-AIM model (http://www.aim.env.uea.ac.uk/aim/aim.php) includes the dissociation equilibrium for some organic/inorganic systems. The oxalic acid is included in current E-AIM. What would E-AIM predict and how does that compare with your experimental data? This can not only serve as validation of the determined Raman growth factor but may also provide additional insides to the effects of reactions on particle's hygroscopicity.**

*Reply: We thank the reviewer for the good suggestion. In fact, the Raman growth factors of pure ammonium sulfate and oxalic acid have been given in Fig. 5 (i.e., Fig. 9 in new version) for comparisons. It is clear that the two species show comparable hygroscopic growth at high RH (dehydration curve, Fig. 5(a)). According to the ZSR rule, the hygroscopic growth of mixtures of ammonium sulfate and oxalic acid should be close to that of pure ammonium sulfate or oxalic acid. Due to lack of Raman cross section data, our Raman growth factors could not be converted into ZSR-predictions. Thus, we used Raman growth factors of pure ammonium sulfate and oxalic acid to compare with that of mixtures in the original manuscript.*

*As for the E-AIM, our previous study by Jing et al. (2016) has shown that E-AIM could well describe the hygroscopic growth of equal mass mixture of ammonium sulfate and oxalic acid, which underwent rapid dehydration in the HTDMA system. As stated in the Discussion section, the HTDMA studies observed no formation of ammonium hydrogen oxalate or influence of interactions between ammonium sulfate and oxalic acid on water uptake of mixtures. Also, the E-AIM does not consider the formation of solid ammonium hydrogen oxalate. As a result, it can be expected that the E-AIM could not well describe water uptake of mixed ammonium sulfate/oxalic acid particles undergoing the slow drying process. This situation also applies to AIOMFAC model.*

**References**

Brooks, S. D., Wise, M. E., Cushing, M., and Tolbert, M. A.: Deliquescence behavior of organic/ammonium sulfate aerosol, Geophys. Res. Lett., 29, 1917, doi: 10.1029/2002gl014733, 2002.

Ghorai, S., Wang, B., Tivanski, A., and Laskin, A.: Hygroscopic properties of internally mixed particles composed of NaCl and water-soluble organic acids, Environ. Sci. Technol., 48, 2234-2241, doi: 10.1021/es404727u, 2014.

Jing, B., Tong, S. R., Liu, Q. F., Li, K., Wang, W. G., Zhang, Y. H., and Ge, M. F.: Hygroscopic behavior of multicomponent organic aerosols and their internal mixtures with ammonium sulfate, Atmos. Chem. Phys., 16, 4101-4118, 2016.

Miñambres, L., Méndez, E., Sánchez, M. N., Castaño, F., and Basterretxea, F. J.: Water uptake of internally mixed ammonium sulfate and dicarboxylic acid particles probed by infrared spectroscopy, Atmos. Environ., 70, 108-116, doi: 10.1016/j.atmosenv.2013.01.007, 2013.

Peng, C. G., Chan, M. N., and Chan, C. K.: The hygroscopic properties of dicarboxylic and multifunctional acids: Measurements and UNIFAC predictions, Environ. Sci. Technol., 35, 4495-4501, doi: 10.1021/es0107531, 2001.

Prenni, A. J., DeMott, P. J., and Kreidenweis, S. M.: Water uptake of internally mixed particles containing ammonium sulfate and dicarboxylic acids, Atmos. Environ., 37, 4243–4251, doi: 10.1016/S1352-2310(03)00559-4, 2003.

Wise, M. E., Surratt, J. D., Curtis, D. B., Shilling, J. E., and Tolbert, M. A.: Hygroscopic growth of ammonium sulfate/dicarboxylic acids, J. Geophys. Res., 108, 4638, doi: 10.1029/2003jd003775, 2003.